# Interneuron FGF13 regulates seizure susceptibility via a sodium channel-independent mechanism

**Susan Lin[1†], Aravind R Gade[1†], Hong-Gang Wang[1], James E Niemeyer[2], Allison Galante[1], Isabella DiStefano[1], Patrick Towers[1], Jorge Nunez[1], Maiko Matsui[1], Theodore H Schwartz[2], Anjali Rajadhyaksha[3,4], Geoffrey S Pitt[1]***

[1]Cardiovascular Research Institute, Weill Cornell Medicine, New York City, United States; [2]Department of Neurological Surgery and Brain and Mind Research Institute, Weill Cornell Medicine of Cornell University, New York Presbyterian Hospital, New York, United States; [3]Department of Pediatrics, Division of Pediatric Neurology, Weill Cornell Medicine, New York City, United States; [4]Brain and Mind Research Institute, Weill Cornell Medicine, New York, United States

**\*For correspondence:**
geoffrey.pitt@med.cornell.edu

[†]These authors contributed equally to this work

## eLife Assessment

This **important** study advances our understanding of how FGF13 variants confer seizure susceptibility. By acting in a set of inhibitory interneurons, FGF13 regulates synaptic transmission and excitability. The data presented here are **convincing** and combine cell type-specific knockouts and electrophysiology, complemented by histology/RNA studies. Collectively, this research will be of interest to a wide audience, particularly those involved in the study of epilepsy, inhibitory neurons, and ion channels.

**Abstract** Developmental and epileptic encephalopathies (DEEs), a class of devastating neurological disorders characterized by recurrent seizures and exacerbated by disruptions to excitatory/inhibitory balance in the brain, are commonly caused by mutations in ion channels. Disruption of, or variants in, *FGF13* were implicated as causal for a set of DEEs, but the underlying mechanisms were clouded because *FGF13* is expressed in both excitatory and inhibitory neurons, *FGF13* undergoes extensive alternative splicing producing multiple isoforms with distinct functions, and the overall roles of FGF13 in neurons are incompletely cataloged. To overcome these challenges, we generated a set of novel cell-type-specific conditional knockout mice. Interneuron-targeted deletion of *Fgf13* led to perinatal mortality associated with extensive seizures and impaired the hippocampal inhibitory/excitatory balance while excitatory neuron-targeted deletion of *Fgf13* caused no detectable seizures and no survival deficits. While best studied as a voltage-gated sodium channel (Na$_v$) regulator, we observed no effect of *Fgf13* ablation in interneurons on Na$_v$s but rather a marked reduction in K$^+$ channel currents. Re-expressing different *Fgf13* splice isoforms could partially rescue deficits in interneuron excitability and restore K$^+$ channel current amplitude. These results enhance our understanding of the molecular mechanisms that drive the pathogenesis of *Fgf13*-related seizures and expand our understanding of FGF13 functions in different neuron subsets.

## Introduction

Developmental and epileptic encephalopathies are a devastating group of rare epilepsies and neurodevelopmental disorders (*Scheffer et al., 2016*; *Scheffer et al., 2017*; *Scheffer and Liao, 2020*) that

affect 4 in 10,000 infants per year (*Covanis, 2012*; *Ware et al., 2019*). DEEs are characterized by repeated seizures, severe developmental delay, and are often drug resistant. Febrile seizures, which are more common in young children, can contribute to the progression of disease (*Skotte et al., 2022*). DEEs are lethal when the seizures lead to sudden unexpected death in epilepsy (SUDEP; *Scheffer and Nabbout, 2019*; *Chilcott et al., 2022*; *Donnan et al., 2023*). Although not all DEEs have been linked to a genetic basis, variants in ion channels and their auxiliary proteins are among the best-characterized genetic causes for DEEs, and sodium channelopathies have been identified as a top cause (*Meisler et al., 2021*).

Epilepsy-associated variants in voltage-gated sodium channels, necessary for action potential initiation, exert gain-of-function effects in excitatory neurons or loss-of-function effects in inhibitory neurons, leading to an overall pro-excitatory state. Gain-of-function variants in *SCN8A* coding for the neuronal sodium channel Na$_V$1.6 (*Adam et al., 1993*; *Bunton-Stasyshyn et al., 2019*; *Meisler, 2019*; *Talwar and Hammer, 2021*) or *SCN2A* coding for Na$_V$1.2 (*Kim et al., 2020*; *Li et al., 2021Miao et al., 2020*) affect excitatory neurons of the forebrain, and lead to network hyperexcitability that drives seizures. In contrast, loss-of-function variants in *SCN1A* coding for Na$_V$1.1, the most abundant Na$_V$ expressed in forebrain inhibitory neurons, reduce inhibitory drive to excitatory neurons and thus drive network hyperexcitability as the mechanism for seizures (*Vormstein-Schneider et al., 2020*; *Ding et al., 2021*). Variants in Na$_V$1.1 underlie the most common genetic subgroup in the prototypical DEE, Dravet Syndrome (*Scheffer and Nabbout, 2019*; *Han et al., 2020*, *Brunklaus et al., 2022*; *He et al., 2022*). DEEs also result from variants in Na$_V$ auxiliary subunits, such as in the Na$_V$ beta subunit *SCN1B* (*Patino et al., 2009*; *Ogiwara et al., 2012*; *Kim et al., 2013*; *Ramadan et al., 2017*; *Bouza and Isom, 2018*; *Hull et al., 2020*) as well as *FGF12* (*Siekierska et al., 2016*; *Oda et al., 2019*; *Saleem et al., 2024*) and *FGF13* (*Puranam et al., 2015*; *Fry et al., 2021*, *Velíšková et al., 2021*; *Narayanan et al., 2022*), both fibroblast growth factor homologous factors (FHFs; *Goldfarb, 2005*).

Here, we focus on *FGF13*, which is expressed in both excitatory and inhibitory neurons (*Puranam et al., 2015*; *Joglekar et al., 2021*). Like other members of the FHF family (FGF11-14), FGF13 is a non-canonical fibroblast growth factor that is not secreted and does not bind to FGF receptors (*Schoorlemmer and Goldfarb, 2001*; *Olsen et al., 2003*). FHFs reside in the cytoplasm where they can bind to the intracellular C-terminus of various Na$_V$s and modulate their function (*Goldfarb, 2005*; *Pablo et al., 2016*). FGF13 has been shown to be upregulated in the hippocampus of a temporal lobe epilepsy mouse model, whereas knockdown of *Fgf13* attenuated hyperexcitability in hippocampal cells (*Shen et al., 2022*). Other FGF13 functions, such as the regulation of microtubules (*Wu et al., 2012*), control of mitogen-activated protein kinases (*Lu et al., 2015*), and effects on neurodevelopment (*Nishimoto and Nishida, 2007*) including development of chandelier cells (*Favuzzi et al., 2019*) have been ascribed, but less well characterized. Moreover, the full complement of FGF13 functions has yet to be defined, obscuring insight into how *FGF13* variants lead to DEEs.

While DEE associated variants in *FGF13* are thought to affect Na$_V$ function, this has not been demonstrated in neurons and the contribution of other FGF13 functions has not been investigated, thus preventing an understanding of disease pathology and the development of precision therapies. Lack of mechanistic clarity also arises because DEE associated variants in *FGF13* have been proposed to cause gain-of-function effects in excitatory neurons (*Fry et al., 2021*) or via an apparently contradictory loss-of-function mechanism in inhibitory neurons (*Puranam et al., 2015*). Further complicating mechanistic understanding, *FGF13* can generate multiple isoforms, driven by alternative promoter usage and consequent alternative splicing, with different but overlapping functions (*Munoz-Sanjuan et al., 2000*; *Puranam et al., 2015*). We recently discovered that alternatively spliced *Fgf13* mRNAs exhibit distinct cell type-specific expression in the mouse cortex. In the developing mouse brain excitatory neurons express almost exclusively *Fgf13-S* while inhibitory neurons express both *Fgf13-S* and *Fgf13-VY*, a notable example of differential expression of an alternatively spliced neuronal gene in excitatory vs. inhibitory neurons (*Furlanis et al., 2019*; *Joglekar et al., 2021*). This cell-type-specific alternative splicing pattern is similar to other genes which are implicated in the control of synaptic interactions in the hippocampus (*Furlanis et al., 2019*), yet the differential splicing complicates the understanding of current models.

DEE-associated variants in *FGF13* have been identified in *FGF13-S* and modeled as causing pro-excitatory Na$_V$ channel dysfunction in excitatory neurons. Conversely, a maternally transmitted chromosomal translocation that disrupts the X-linked *FGF13* locus and eliminates *FGF13-VY* expression

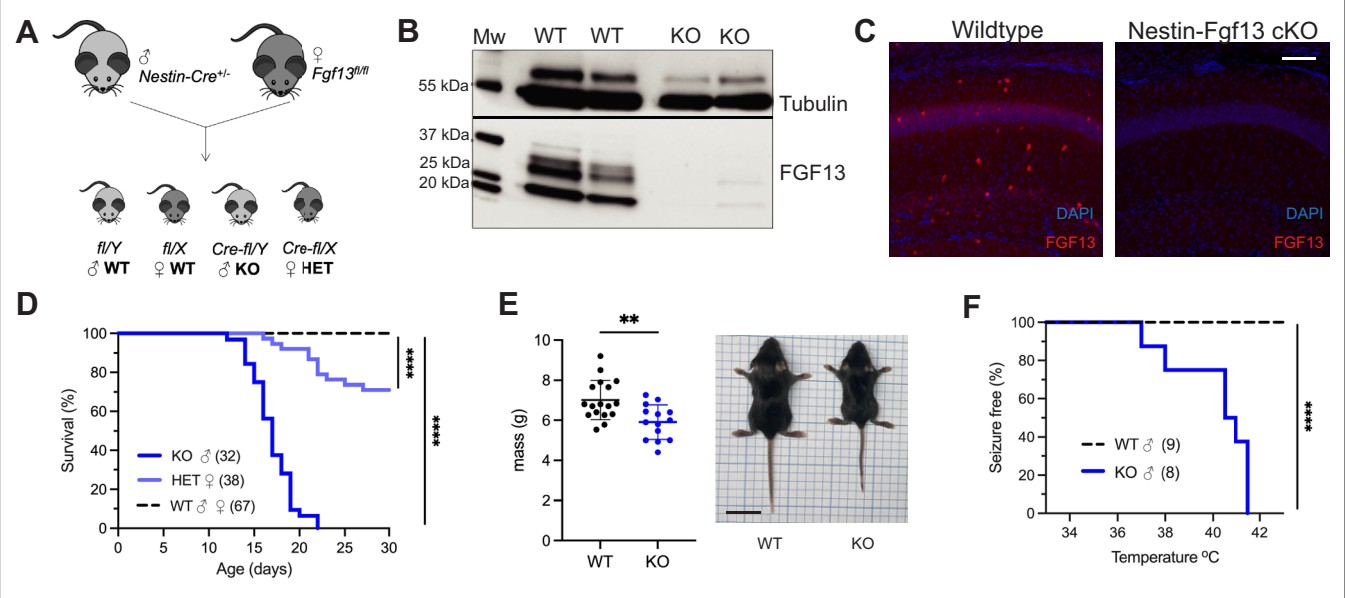

**Figure 1.** Whole brain knockout of *Fgf13* results in premature death and seizure susceptibility. (**A**) Breeding scheme to generate Nestin-Fgf13 cKO, Nestin-Fgf13 Het, and wildtype littermates. (**B**) Western blots of whole brain Nestin-Fgf13 cKO (KO) and wildtype (WT) littermates at P2 validates *Fgf13* knockout. Tubulin used as a loading control. (**C**) Fluorescent immunohistochemistry of hippocampal tissue validates *Fgf13* knockout (scale bar, 100 µm). (**D**) Survival curve of Nestin-Fgf13 cKO mutant mice shows decreased survival at 1 month of age (log-rank test, ****, p<0.0001). (**E**) Body mass at P14 shows Nestin-Fgf13 cKO are smaller in size (t-test, **, p<0.01). (**F**) Nestin-Fgf13 cKO are susceptible to hyperthermia-induced seizures (log-rank test, ****, p<0.0001), unlike wildtype littermates.

The online version of this article includes the following source data and figure supplement(s) for figure 1:

**Source data 1.** Original TIFF files saved from BioRad Gel doc system for gels shown in *Figure 1B*, together with a PDF file identifying the respective portions displayed.*Figure 1B*.

**Figure supplement 1.** *Fgf13* splice variants are differentially expressed in hippocampal cell types.

**Figure supplement 1—source data 1.** Original TIFF files saved from BioRad Gel doc system for gels shown in *Figure 1—figure supplement 1B*, together with a PDF file identifying the respective portions displayed.

while preserving expression of *FGF13-S* causes a seizure disorder in two brothers (*Puranam et al., 2015*). Thus, how *FGF13* variants confer seizure susceptibility is unclear. Finally, whether *FGF13* variants even exert their effects through Na$_V$ dysfunction or through alternative mechanisms has not been defined.

We generated a series of conditional knockout mouse lines to probe FGF13 function in excitatory and inhibitory hippocampal neurons and investigated the isoform-specific and cell type mechanisms by which variants in *Fgf13* cause DEEs. We define a critical role for FGF13 in interneurons but not excitatory neurons underlying a pro-excitatory state associated with seizures. Moreover, as we observe limited effects on Na$_V$ currents underlying the changes associated with a pro-excitatory state, our data add to growing observations that FGF13 can function independently of Na$_V$s.

## Results

### Whole brain knockout of *Fgf13* results in spontaneous seizures and premature death

To determine how *FGF13* affects neuronal excitability and provide insight into how *FGF13* variants contributes to seizure disorders, we developed genetic mouse models that eliminate *Fgf13* in specific neuronal cell types. We started by eliminating the X-linked *Fgf13* in all neurons, crossing a male *Nestin*-Cre driver with a *Fgf13$^{fl/fl}$* female mouse from a previously validated line (*Wang et al., 2017*; *Figure 1A*) to generate hemizygous male *Fgf13* knockouts (*Nes-cre;Fgf13$^{fl/Y}$* or 'Nestin-Fgf13 cKO') and heterozygous females (*Fgf13$^{fl/+}$* or 'Nestin-Fgf13 Het') alongside wild type (WT) littermates. We validated our mouse lines by examining the hippocampus, where *Fgf13* is highly expressed (*Pablo et al., 2016*) and

**Table 1.** Mutant mice are born in Mendelian ratios.

| Mouse strain | Probability density function |
|---|---|
| Nestin-Fgf13 | 0.23 |
| Gad2-Fgf13 | 0.22 |
| Nkx2.1-Fgf13 | 0.24 |
| Emx1-Fgf13 | 0.21 |

Mutant mice were born in expected Mendelian ratios. Despite embryonic targeted deletion of *Fgf13,* mutant suffered no prenatal mortality related to developmental deficits.

which has previously been implicated in *FGF13*-related seizure pathology (*Puranam et al., 2015*; *Shen et al., 2022*) and cognition (*Wu et al., 2012*). In hippocampus from WT mice, western blotting with a pan-FGF13 antibody that recognizes all FGF13 isoforms detected four distinct bands that we were able to assign to specific FGF13 isoforms by running standards (isoform specific cDNAs expressed in HEK293 cells; *Figure 1*, *Figure 1—figure supplement 1*). In hippocampi from Nestin-Fgf13 cKO brains we observed near complete absence of FGF13 (*Figure 1B*) by western blot and by immunohistochemistry (*Figure 1C*). In WT mice, FGF13 signal is prominent in the hippocampal pyramidal cell layer as well as in sparse presumed interneuron somata (see below) throughout all layers. Offspring were born at expected Mendelian ratios (*Table 1*), but all Nestin-Fgf13 cKO male mice died before weaning with a median survival age of 17 days postnatal (P17; *Figure 1D*). Nestin-Fgf13 Het females also exhibited postnatal death, with only 60% surviving past 1 month (*Figure 1D*), implying a gene dosage effect of *Fgf13* loss. Nestin-Fgf13 cKO male mice suffered spontaneous seizures in their home cage (*Video 1*) as early as P12, followed by premature death (*Figure 1D*), consistent with mouse models of DEE that exhibit SUDEP (*Bunton-Stasyshyn et al., 2019*; *Han et al., 2020*). Nestin-Fgf13 cKO males were smaller than their wild type littermates by P14 (*Figure 1E*), suggesting developmental delay, likely a consequence of recurrent seizures and consistent with DEE mouse models (*Yu et al., 2006*; *Kalume et al., 2013*; *Teran et al., 2023*).

As febrile seizures are a hallmark of early developmental onset epilepsy and were observed in subjects with a gene translocation that disrupted the *FGF13* locus (*Puranam et al., 2015*), we tested hyperthermia-induced seizure susceptibility in the Nestin-Fgf13 cKO male mice, following a modified protocol (*Cheah et al., 2012*; *Puranam et al., 2015*). Mice were tested at P12, the age at which we first observed spontaneous home cage seizures. Briefly, mice were acclimated to the testing chamber and a heat lamp was used to increase core body temperature at a steady rate of 0.5 °C every 2 min until 42.0 °C to simulate fever onset. All Nestin-Fgf13 cKO males exhibited tonic-clonic seizures during the hyperthermic seizure protocol while wild type mice remained seizure free (*Figure 1F*).

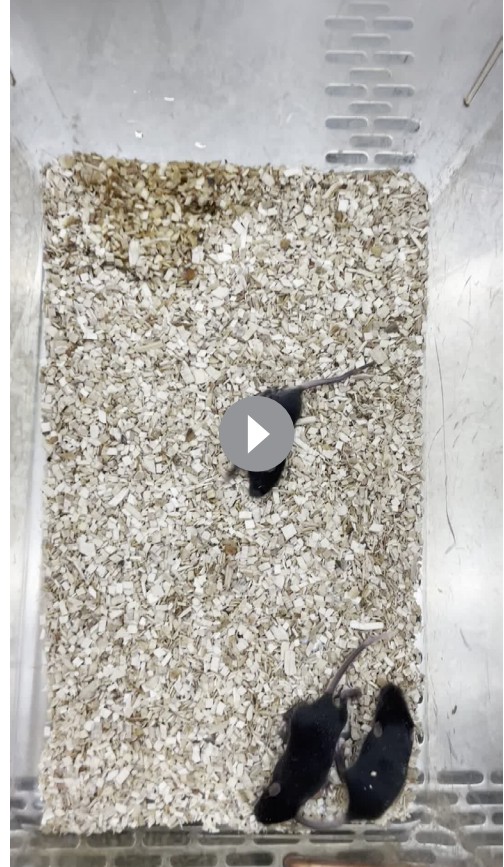

**Video 1.** Nestin-Fgf13 cKO male mice suffered spontaneous seizures in their home cage. Example spontaneous seizure activity is shown.

https://elifesciences.org/articles/98661/figures#video1

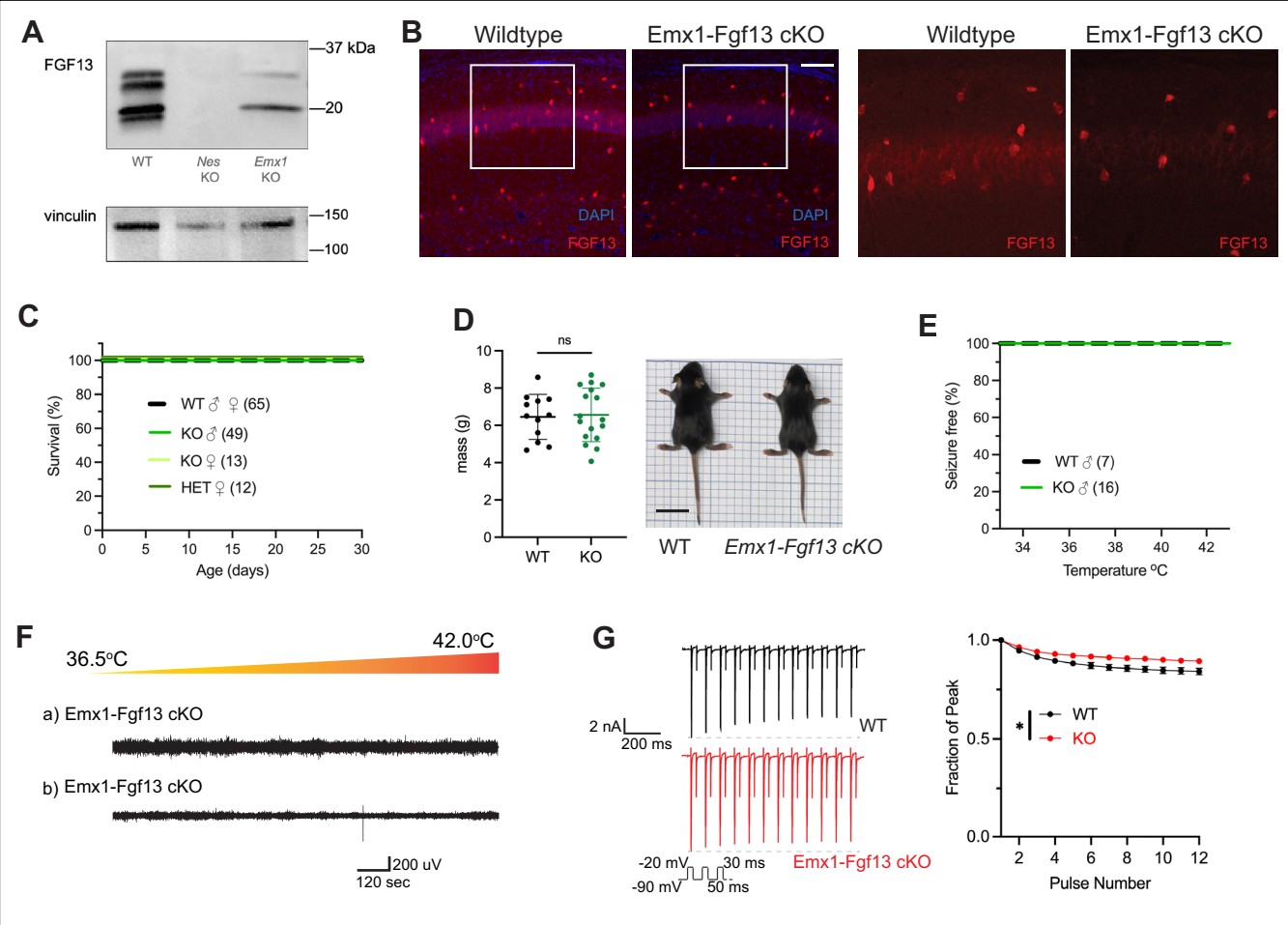

**Figure 2.** Excitatory neuronal knockout of *Fgf13* does not result in premature death and seizure susceptibility. (**A**) Western blot shows partial loss of Fgf13 from Emx1-Fgf13 cKO hippocampal tissue, compared to full knockout in Nestin-Fgf13 cKO hippocampus. Vinculin used as a loading control. (**B**) Fluorescent immunohistochemistry of hippocampal tissue validates *Fgf13* knockout (scale bar, 100 μm). (**C**) Emx1-Fgf13 cKO mutant mice survive past 1 month of age (log-rank test, p=ns). (**D**). Body mass at postnatal day 14 (P14) shows that Emx1-Fgf13 cKO are not different in size (scale bar, 2 cm) (t-test, p=ns). (**E**) Emx1-Fgf13 cKO are not susceptible to hyperthermia induced seizures (log-rank test, p=ns). (**F**) EEG recordings during hyperthermia protocol show Emx1-Fgf13 cKO do not exhibit heat-induced seizures. (**G**) Emx1-Fgf13 cKO neurons exhibit diminished long-term inactivation (two-way ANOVA, *, p<0.05), though the deficit is not sufficient to cause seizures (WT, N=2, n=15; KO, N=2, n=15). Example traces for WT and Emx1-Fgf13 cKO neurons are shown on the left.

The online version of this article includes the following source data and figure supplement(s) for figure 2:

**Source data 1.** Original TIFF files saved from Bio-Rad Gel doc system for gels shown in *Figure 2A*, together with a PDF file identifying the respective portions displayed.

**Figure supplement 1.** Excitatory neuronal knockout of *Fgf13* results in loss of FGF13-S.

**Figure supplement 1—source data 1.** Original TIFF files saved from Bio-Rad Gel doc system for gels shown in *Figure 2—figure supplement 1*, together with a PDF file identifying the respective portions displayed.

**Figure supplement 2.** Excitatory neuronal knockout of *Fgf13* does not result in sodium current deficits.

## Excitatory neuronal knockout of *Fgf13* does not result in spontaneous seizures

Excitatory neurons have been the focus of most studies investigating *Fgf13* dysfunction. *Fgf13* deficiency in excitatory forebrain neurons is linked to neurodevelopmental delay and cognitive impairment (*Wu et al., 2012*), and excitatory neurons are hypothesized to be the relevant cell type for seizures in a DEE associated with *FGF13* variants (*Fry et al., 2021*). To test the contribution of excitatory neurons to the seizure and SUDEP phenotype identified in Nestin-Fgf13 cKO mice, we generated an excitatory neuron-targeted knockout mouse using an *Emx1*-Cre driver (*Gorski et al., 2002*).

We assessed efficacy of knockout by western blot using the pan-FGF13 antibody. In Emx1-Fgf13 cKO mice, western blots showed the loss of two FGF13 isoforms (*Figure 2A*), FGF13-S and FGF13-U, consistent with single-cell long-read RNA sequencing that showed that excitatory neurons predominantly express FGF13-S (*Joglekar et al., 2021*), which we confirmed with a FGF13-S specific antibody (*Figure 2—figure supplement 1*). FGF13-VY and FGF13-V, the major isoforms expressed in inhibitory interneurons (*Joglekar et al., 2021*), were preserved. Thus, these data confirm at the protein level the differential expression of specific *Fgf13* transcripts in excitatory vs. inhibitory neurons observed by single cell RNA sequencing. Fluorescent immunohistochemistry of hippocampi from Emx1-Fgf13 cKO showed loss of FGF13 from the CA1 pyramidal cell layer but maintenance of FGF13 signal from sparse putative interneuron somata (*Figure 2B*). Emx1-Fgf13 cKO mice were born at expected Mendelian ratios (*Table 1*) and all survived to adulthood (*Figure 2C*), similar to a previous report (*Wu et al., 2012*). Average body size and mass at P14 revealed no differences between wild type and Emx1-Fgf13 cKO genotypes (*Figure 2D*). None of the Emx1-Fgf13 cKO succumbed to seizures during the hyperthermia protocol (*Figure 2E*), and EEG recordings during the hyperthermia protocol confirmed no seizure activity in Emx1-Fgf13 cKO or wild type mice (*Figure 2F*). These data show that loss of *Fgf13* from excitatory forebrain neurons is not sufficient to induce seizure susceptibility.

The mechanism by which DEE-associated variants in *FGF13-S* contribute to a pro-excitatory state and seizures was hypothesized to be loss of a FGF13-S-dependent long-term inactivation, a process that describes a successive reduction in peak $Na_V$ current following repetitive depolarizations and is thought to derive from accreting channel pore blockade by the alternatively spliced N-terminus of FGF13-S (*Fry et al., 2021*). Whether FGF13-S confers long-term inactivation in neurons has not been demonstrated. We employed the knockout model to assess the effects of FGF13-S on $Na_V$ currents in *Emx1+* excitatory neurons and determine if FGF13-S bestows long-term inactivation. We cultured hippocampal neurons from Emx1-Fgf13 cKO or wild type littermates (*Emx-Cre*) and infected them with a Cre-dependent AAV8-DIO-GFP virus to allow identification of the *Emx1+* neurons by GFP (*Figure 2—figure supplement 2A*). The major neuronal voltage gated sodium channels ($Na_V1.1$, $Na_V1.2$, $Na_V1.3$, and $Na_V1.6$) are expressed in these neuron cultures consistent with levels previously reported (*Heighway et al., 2022*), as assessed by reverse transcriptase quantitative polymerase chain reaction (*Figure 2—figure supplement 2B*). Current density and the I-V relationship were not different between WT and Emx1-Fgf13 cKO neurons (*Figure 2—figure supplement 2C*), and the $V_{1/2}$ of steady-state inactivation was not significantly different in Emx1-Fgf13 cKO neurons compared to WT controls (*Figure 2—figure supplement 2D*). To assess for long-term inactivation, we applied successive depolarizations (*Figure 2G*) and observed successive reductions in peak current in WT neurons, a process blunted in Emx1-Fgf13 cKO neurons. Thus, although *Fgf13* ablation in *Emx1+* neurons diminished long-term inactivation—similar to that observed for DEE associated variants in *FGF13-S* expressed in Neuro2A cells (*Fry et al., 2021*)—these data suggest that FGF13-S dysfunction in excitatory neurons is an unlikely contributor to a pro-excitatory state and seizures.

## Inhibitory neuron knockout of *Fgf13* recapitulates spontaneous seizures and premature death in complete neuronal knockout

Since *Fgf13* knockout in excitatory neurons did not recapitulate the seizure phenotype seen in the Nestin-Fgf13 cKO animals, we suspected that absence of *Fgf13* specifically in inhibitory interneurons may be playing the critical role. With immunohistochemistry on brains from mice in which a *Gad2*-Cre driver (*Taniguchi et al., 2011*) expressed a Cre-dependent GFP (Ai6) reporter (*Madisen et al., 2010*) in inhibitory interneurons (*Figure 3—figure supplement 1A–B*), one-third of the GFP+ interneurons expressed FGF13 at P13 (*Figure 3A*). We eliminated *Fgf13* specifically in inhibitory neurons by crossing the *Gad2-Cre* driver and *Fgf13fl/fl* mice. The FGF13 signal by immunohistochemistry was completely eliminated from the GFP+ cells in mouse brains also expressing GFP from the *Gad2-Cre* driver while the diffuse staining pattern in the excitatory pyramidal cell layer and overall hippocampal morphology were unaffected (*Figure 3—figure supplement 1A–B*). Further, higher magnification images showed that in Gad2-Fgf13 cKO mice the FGF13+ staining in sparse somata (presumed interneurons above) was completely eliminated (*Figure 3B*). Western blots with the pan-FGF13 antibody showed partial depletion of FGF13 in Gad2-Fgf13 cKO (*Figure 3C*), with the most significant reduction in FGF13-VY and FGF13-Y while FGF13-S was partially spared, consistent with our analysis of splice-variant specific transcript data (*Joglekar et al., 2021*).

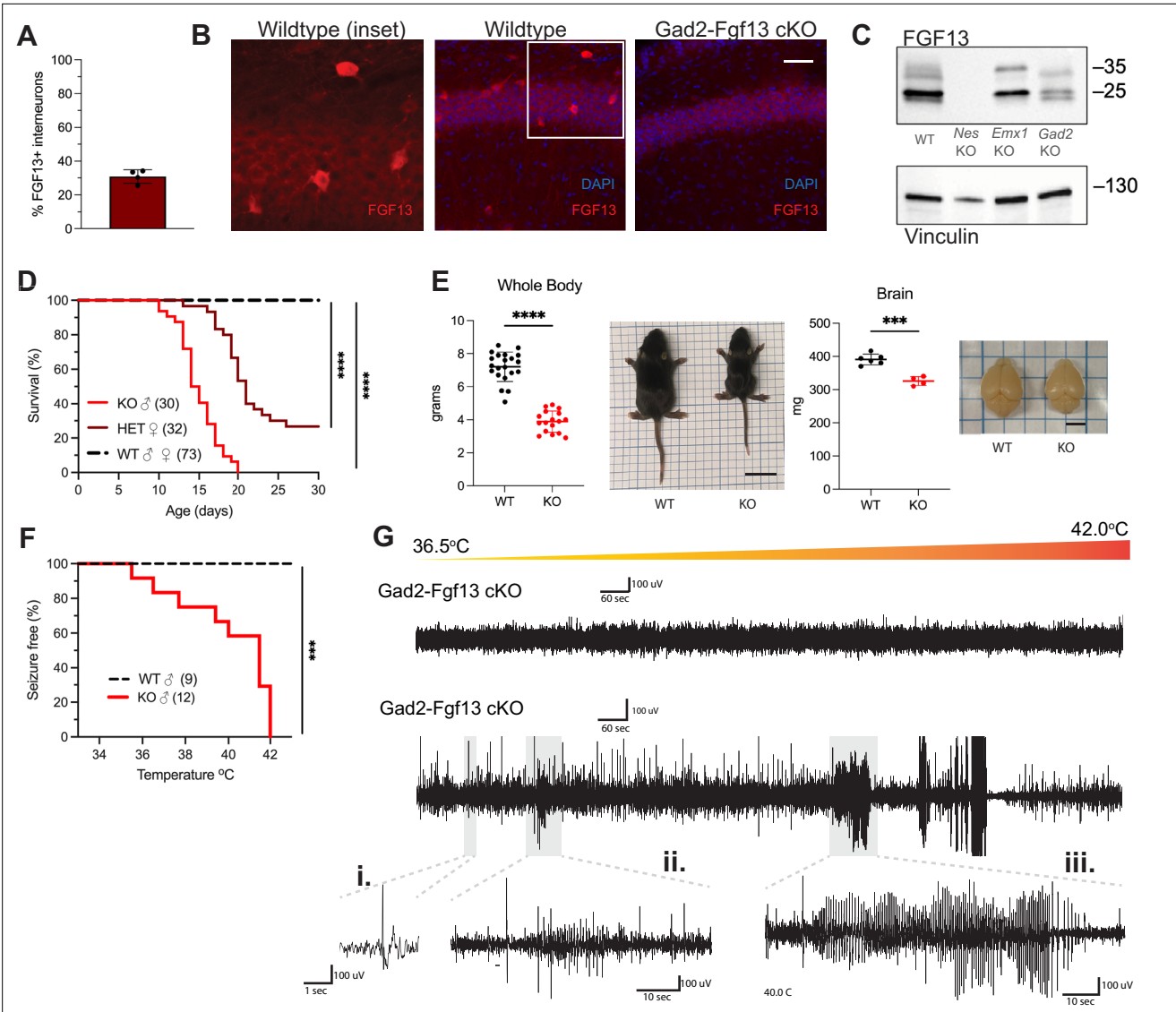

**Figure 3.** Inhibitory neuronal knockout of *Fgf13* recapitulates premature death and seizure susceptibility. (**A**) Quantification of hippocampal interneuron histology reveals 31% of *Gad2*[+] interneurons co-express FGF13. (**B**) Fluorescent immunohistochemistry of hippocampal tissue validates *Fgf13* knockout in sparse inhibitory interneurons (scale bar, 50 μm). (**C**) Western blot validates partial loss of Fgf13 from Emx1-Fgf13 cKO and Gad2-Fgf13 cKO hippocampal tissue, and full knockout in Nestin-Fgf13 cKO mice. Vinculin used as a loading control. (**D**) Gad2-Fgf13 cKO mutant mice have survival deficits around 1 month of age (log-rank test, ****, p<0.0001). (**E**) Body mass at P14 shows Gad2-Fgf13 cKO are smaller in size than wildtype littermates (t-test, ****, p<0.001; scale bar, 2 cm) and brains from Gad2-Fgf13 cKO mice are smaller (t-test, ***, p<0.001; scale bar, 0.5 cm). (**F**) Gad2-Fgf13 cKO mice are susceptible to hyperthermia induced seizures (log-rank test, ***, p<0.001). (**G**) Example EEG recording of wildtype and Gad2-Fgf13 cKO mice during hyperthermia protocol. Insets show (**i**) example interictal spike, (**ii**) and (**iii**) tonic-clonic seizures.

The online version of this article includes the following source data and figure supplement(s) for figure 3:

**Source data 1.** Original TIFF files saved from Bio-Rad Gel doc system for gels shown in *Figure 3C* and PDF, together with a identifying the respective portions displayed.

**Figure supplement 1.** Characterization of FGF13 abundance in hippocampus of Gad2-Fgf13 cKO mice.

**Figure supplement 2.** *Fgf13* splice variants are differentially expressed in hippocampal cell types.

**Figure supplement 3.** An unbiased metabolomic analysis on whole brain lysate from Gad2-Fgf13 cKO and wildtype mice reveals marked differences between genotypes.

**Figure supplement 4.** Knockout of *Fgf13* from MGE-derived interneurons recapitulates premature death and seizure susceptibility.

In conjunction with the generation of the Nestin-Fgf13 cKO and Emx1-Fgf13 cKO mice, the Gad2-Fgf13 cKO mice provided the opportunity to validate and extend the previously published transcript data (*Joglekar et al., 2021*) and the protein (western blots in this study) data in cell-type-specific knockout models. Using BaseScope in situ hybridization we characterized *Fgf13-S* and *Fgf13-VY* expression in intact hippocampi on fixed frozen tissue. We detected diffuse *Fgf13-S* expression in excitatory cell layers while *Fgf13-VY* expression was concentrated in interneuron somata in wild type hippocampi (*Figure 3—figure supplement 2A–B*). Quantification of the mRNA signal intensity revealed a decrease in *Fgf13-S* for genotypes in which we deleted *Fgf13* in excitatory cells (Nestin-Fgf13 cKO and Emx1-Fgf13 cKO); meanwhile quantification of the cell somata expressing *Fgf13-VY* revealed a decrease in the *Fgf13-VY*+ cell count for genotypes in which we deleted *Fgf13* in interneurons (Nestin-Fgf13 cKO and Gad2-Fgf13 cKO), as shown in *Figure 3—figure supplement 2C*. In situ hybridization in these knockout models confirms the splice variant-specific transcript data previously obtained (*Joglekar et al., 2021*), although we did not detect the expected decrease in the *Fgf13-S* signal in Gad2-Fgf13 cKO mice, likely because we lacked the resolution due to the over-whelming *Fgf13-S* signal from excitatory neurons.

With this confirmation, we investigated the consequences of *Fgf13* loss from interneurons. None of the male Gad2-Fgf13 cKO hemizygous knockout mice survived to adulthood (*Figure 3D*), similar to the Nestin-Fgf13 cKO male mice. The median survival in male Gad2-Fgf13 cKO mice was 14.5 days, and survivors at P14 were smaller than their wild type littermates (*Figure 3E*). Female Gad2-Fgf13 Het mice also had a survival deficit (*Figure 3D*), though less severe than their male knockout littermates, suggesting a gene dosage effect. Brains of male Gad2-Fgf13 cKO mice were smaller compared to wild type littermates, as was body size (*Figure 3E*).

Interneuron-specific knockout of *Fgf13* increased susceptibility to seizures in the hyperthermia-induced seizure protocol. All Gad2-Fgf13 cKO males suffered tonic-clonic seizures while wild type littermates did not (*Figure 3F–G*). Furthermore, we observed some Gad2-Fgf13 cKO males were susceptible to seizures during acclimation to the testing chamber, prior to the initiation of the hyper-thermia protocol. With witnessed seizures immediately preceding death in some mice, we interpret that the previously noted unwitnessed premature deaths in the home cage were likely related to recurrent seizures. Together, these data suggest that FGF13 deficiency in inhibitory cells, rather than excitatory cells, is the major contributor to the seizures and premature mortality seen in the Nestin-Fgf13 cKO mice.

Given the seizures and mortality in the Gad2-Fgf13 cKO mice and because seizures can be asso-ciated with neuronal death (*Henshall and Simon, 2005*), we considered if *Fgf13* ablation led to loss of hippocampal interneurons. We quantified Gad2-GFP+interneurons in Gad2-GFP (Gad2-GFP vs. Gad2-GFP-Fgf13 cKO) mice, in which Gad2 drove both Fgf13 ablation and expression of a GFP reporter. We found no difference in the number of Gad2-GFP+hippocampal neurons between geno-types (*Figure 3—figure supplement 1C*), indicating that Gad2-Fgf13 cKO mice did not have a deficit of interneurons underlying their seizure phenotype. Further, an unbiased metabolomic analysis on brains from Gad2-Fgf13 cKO and WT control mice revealed marked differences between genotypes (*Figure 3—figure supplement 3* and *Supplementary file 2*) and several features consistent with severe seizures. For example, the metabolite most enriched in Gad2-Fgf13 cKO brains was L-ky-nurenine (Log$_2$fold change = 2.58, p=0.0009), a metabolite most enriched in patients with status epilepticus (*Hanin et al., 2024*). Metabolite set enrichment analysis in Gad2-Fgf13 cKO vs. WT revealed that the most enriched Kyoto Encyclopedia of Genes and Genomes (KEGG) pathway among upregulated metabolites was glycolysis (p=4.29E-6), also consistent with data obtained in patients in status epilepticus (*Hanin et al., 2024*). The most downregulated pathway was methionine metabolism (p=9.88E-4), consistent with methionine synthase deficiency—an inborn error of metabolism—that presents with seizures (*Kripps et al., 2022*). Thus, this metabolomic analysis supported the severity of the seizure phenotype observed in Gad2-Fgf13 cKO mice.

## MGE-derived interneuron knockout of *Fgf13* partially recapitulates spontaneous seizures and premature death

As not all Gad2+ interneurons express *Fgf13* (*Figure 3—figure supplement 1A*), we analyzed a single-cell RNA-seq dataset from mouse CA1 interneurons (*Harris et al., 2018*) to search for a subpopulation functionally relevant to the seizure pathology. We found the highest enrichment of *Fgf13* in subsets

assigned as axo-axonic and medial ganglionic eminence (MGE) neurogliaform/Ivy cells (*Figure 3—figure supplement 4A–B*) expressing the homeodomain transcription factor *Nkx2.1*. Further, *Fgf13* expression has previously been identified in *Nkx2.1*[+] MGE-derived interneurons, including axo-axonic chandelier cells (*Paul et al., 2017*; *Mahadevan et al., 2021*), which depend upon *Fgf13* expression for proper development (*Favuzzi et al., 2019*). Single-cell profiling of human postmortem cortical tissue has identified high *FGF13* expression in neurogliaform and parvalbumin interneurons throughout early development (*Herring et al., 2022*). While *Fgf13* expression was not completely restricted to *Nkx2.1*[+] MGE-derived interneurons, we thus chose an *Nkx2.1*-Cre driver (*Xu et al., 2008*) to target *Fgf13* deletion from interneurons generated in the MGE.

Fluorescent immunohistochemistry in brain slices from Nkx2.1-Fgf13 cKO mice showed partial depletion of the sparse somatic distribution of FGF13[+] somata observed in the Gad2-Fgf13 cKO mice (*Figure 3—figure supplement 4C*). Consistent with that result, BaseScope in situ hybridization in Nkx2.1-Fgf13 cKO mice showed a pattern similar to Gad2-Fgf13 cKO mice but with less complete loss of the interneuron somata (*Figure 3—figure supplement 2B*). Nkx2.1-Fgf13 cKO mice were born at expected Mendelian ratios (*Table 1*) but only ~40% of male Nkx2.1-Fgf13 cKO mice survived past 1 month of age (*Figure 3—figure supplement 4D*), consistent with a partial loss of FGF13 expression in hippocampal interneurons (*Figure 3—figure supplement 4C*). The median survival age for male Nkx2.1-Fgf13 cKO mice was 22 days (P22), and surviving mutants were smaller than their wild type littermates at P14 (*Figure 3—figure supplement 4E*). Female Nkx2.1-Fgf13 Het survived to adulthood. We found that almost all male Nkx2.1-Fgf13 cKO suffered tonic-clonic seizures during the hyperthermia induction protocol, while wild type littermates did not (*Figure 3—figure supplement 4F*). Because the lethality and seizure susceptibility in the Nkx2.1-Fgf13 cKO mice were reduced compared to the Gad2-Fgf13 cKO mice, we hypothesize that MGE-derived neurons are partially responsible for the phenotype observed in Gad2-Fgf13 cKO mice. The loss of FGF13 from other interneuron classes in which FGF13 is expressed—as evident from single cell RNA sequencing (*Figure 3—figure supplement 4A–B*)—likely contributes to the more severe phenotype after broader inhibitory interneuron knockout by the *Gad2*-Cre driver.

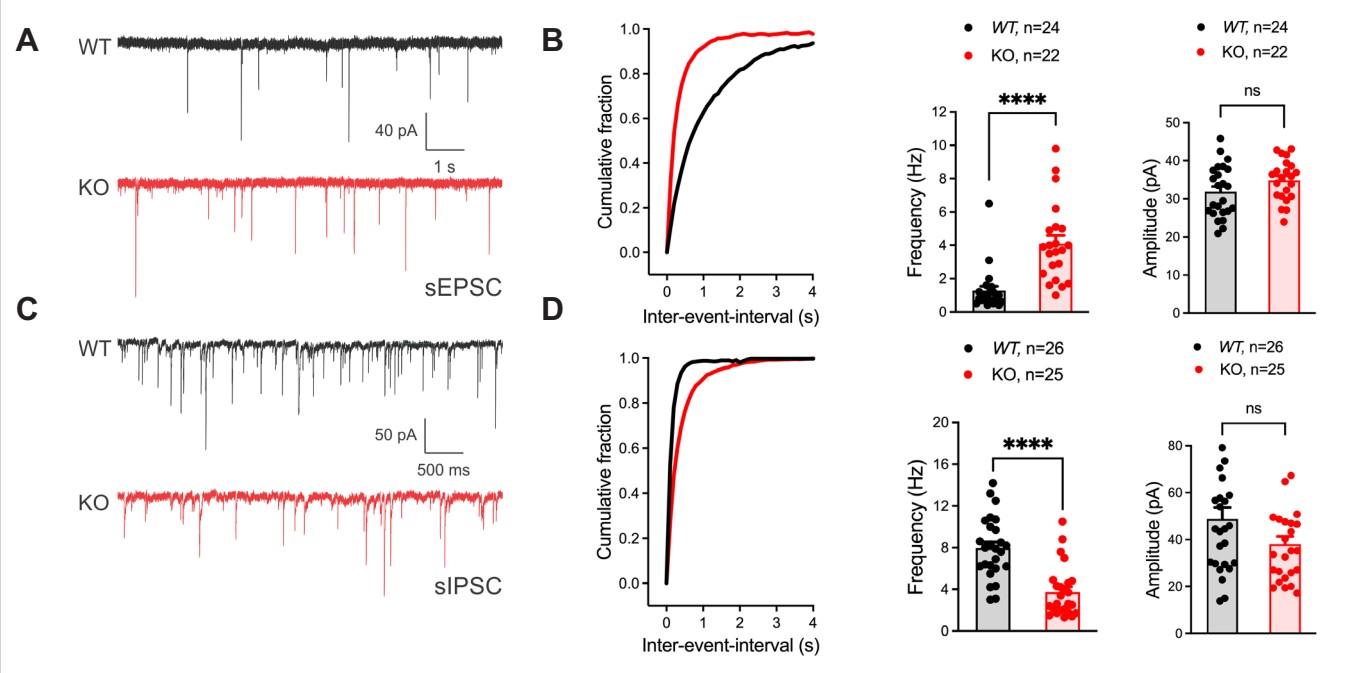

**Figure 4.** Inhibitory neuronal knockout of *Fgf13* results in deficits of synaptic transmission. (**A**) Example traces from spontaneous excitatory postsynaptic currents (sEPSCs). (**B**) Cumulative fraction (Kolmogorov Smirnov test, ****, p<0.0001) frequency, and current amplitude for sEPSCs (t-test, ****, p<0.0001, WT N=7, n=24; KO N=4, n=22.) (**C**) Example traces from spontaneous inhibitory postsynaptic currents (sIPSCs). (**D**) Cumulative fraction (Kolmogorov Smirnov test, ****, p<0.0001), frequency, and current amplitude for sIPSCs (t-test, ****, p<0.0001, WT N=3, n=26; KO N=4, n=25).

## Inhibitory neuronal knockout of *Fgf13* results in deficits of synaptic transmission

To identify if the seizure susceptibility of the Gad2-Fgf13 cKO mice is caused by loss of excitatory/inhibitory balance in the hippocampus, we recorded spontaneous excitatory and inhibitory inputs to the CA1 pyramidal cells of the dorsal hippocampus in slice recordings. Pyramidal cells in Gad2-Fgf13 cKO hippocampi, compared to wild type littermates, showed an increased frequency of sEPSCs (*Figure 4A–B*) and a decreased frequency of sIPSCs mice relative to wild type littermates (*Figure 4C–D*). Because there was no significant change in sEPSC or sIPSC amplitude (*Figure 4B and D*), the increased sEPSC and decreased sIPSC frequencies indicate a presynaptic and not postsynaptic mechanism. Together, these data suggest that ablation of *Fgf13* specifically in inhibitory neurons (in Gad2-Fgf13 cKO mice) decreased direct and downstream inhibitory drive onto pyramidal cells, thereby increasing pyramidal cell excitability.

## Inhibitory neuronal knockout of *Fgf13* results in deficits of interneuron excitability

To assess the cellular mechanisms underlying decreased inhibitory drive from the inhibitory neurons in Gad2-Fgf13 cKO mice, we cultured hippocampal neurons from Gad2-Fgf13 cKO or wild type littermates (*Gad2-Cre*) and infected them with a Cre-dependent AAV8-DIO-GFP virus, thus allowing identification of the $Gad2^+$ interneurons by GFP (*Figure 5A*). We quantified trains of action potentials in response to current injection in the $GFP^+$ neurons to assess intrinsic excitability. Increasing current injection augmented the number of spikes in WT (*Gad2-Cre*) neurons until a plateau at ~260 pA (*Figure 5B*). At low levels (<100 pA) current injection elicited a similar number of spikes in neurons from Gad2-Fgf13 cKO mice (albeit spike morphology and frequency were different, see below) but the number of spikes then plateaued and was less than the number elicited in neurons from WT mice at higher amounts of current injection (*Figure 5B*), suggesting a decrease in intrinsic excitability in neurons from Gad2-Fgf13 cKO mice. Examination of individual traces revealed that more Gad2-Fgf13 cKO cells entered depolarization block, and at lower levels of current injection than wildtype interneurons (*Figure 5C*). The detected difference between genotypes is striking considering that only a third of the $Gad2^+$ neurons from Gad2-Fgf13 cKO infected with the Cre virus would be expected to express FGF13 before viral Cre expression (*Figure 3A*).

To further investigate the mechanisms contributing to decreased excitability and depolarization block, we examined the first elicited action potential properties for each spike train at rheobase. Resting membrane potential, rheobase, and threshold potential were not different between genotypes, *Figure 5D*. The single action potential wave forms and phase plots show little difference between Gad2-Fgf13 cKO neurons infected with the Cre virus and wild type controls (*Figure 5E*), although action potential duration measured at 50% of the amplitude (APD50) trended longer in the Gad2-Fgf13 cKO interneurons infected with the Cre virus (*Figure 5—figure supplement 1A*). However, we observed marked genotype differences in the second and third elicited action potentials. Most notably, in Gad2-Fgf13 cKO interneurons we observed a progressive reduction in the action potential peak amplitude and dV/dt max, an increase in APD50, and a greater increase in the interspike membrane voltage (*Figure 5F–G*), suggesting a growing repolarization deficit in the Gad2-Fgf13 cKO neurons each subsequent spike, thus increasing the likelihood to undergo depolarization block.

Since the first spike action potential parameters were not significantly different between Gad2-Fgf13 cKO neurons and wild type controls, we hypothesized that loss of FGF13 had minimal effects upon $Na_V$ channel currents in these neurons. Indeed, we observed no differences in current amplitude nor steady-state inactivation between Gad2-Fgf13 cKO neurons and wild type controls (*Figure 5—figure supplement 1B*, **C**). Because we did not detect significant differences in $Na_V$ current properties typically associated with FGF13 in *Fgf13* knockout in Gad2-Fgf13 cKO neurons, we hypothesized that FGF13 regulated other currents that led to the observed changes in action potentials and decrease in repolarization. Since in cardiomyocytes we showed that FGF13 can also affect various $K_V$ channels that contribute to the repolarizing phase of the cardiac action potential (*Wang et al., 2017*), here we recorded macroscopic potassium currents and observed a reduced Kv current density in Gad2-Fgf13 cKO interneurons compared to WT neurons (*Figure 5I*).

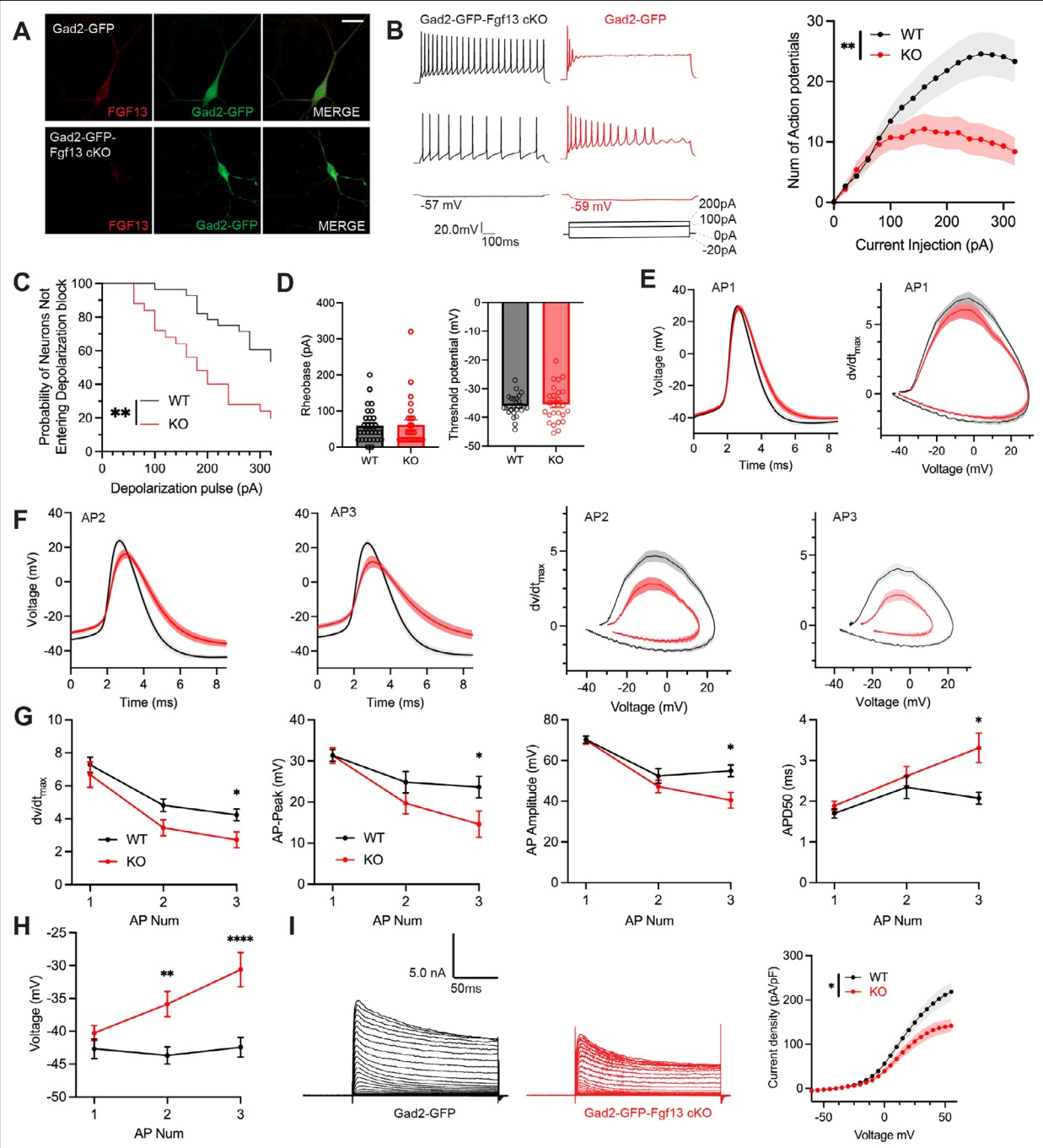

**Figure 5.** Inhibitory neuronal knockout of *Fgf13* results in deficits of interneuron excitability. (**A**) Examples of FGF13-stained neurons from primary hippocampal neuron cultures generated from *Gad2*-Cre wildtype (top) and Gad2-Fgf13 cKO (bottom) male mice (scale bar, 20 μm). (**B**) Example traces of AP spike trains from wildtype (left) and Gad2-Fgf13 cKO interneurons. Input-output curve shows decreased firing of evoked action potentials from Gad2-Fgf13 cKO interneurons (two-way ANOVA, **, p<0.01, WT N=4, n=28; KO N=4, n=26). (**C**) Gad2-Fgf13 cKO interneurons enter depolarization block at earlier current injections than wildtype interneurons (log-rank test, **, p<0.01). (**D**) Resting membrane potential, rheobase, and threshold potential for cultured wildtype and Gad2-Fgf13 cKO interneurons. (**E**) Action potential wave forms and phase plots (mean ± s.e.m.) for the first elicited action potential of the spike train for wildtype and Gad2-Fgf13 cKO interneurons. (**F**) Action potential wave forms and phase plots for the second and third action potentials of the spike train of wildtype and Gad2-Fgf13 cKO interneurons. (**G**) Analysis of the first three action potentials in the spike train show a decrease in the dV/dt max, a decrease in action potential (AP) peak, a decrease in AP amplitude, and an increase in APD50 by the third AP of Gad2-Fgf13 cKO interneurons (two-way ANOVA, *, p<0.05). (**H**) Analysis of the first three action potentials in the spike train show an increase in membrane potential at the end of the AP for Gad2-Fgf13 cKO interneurons (two-way ANOVA, **, p<0.01, ****, p<0.0001). (**I**) Example traces of K⁺

Figure 5 continued

currents from wildtype and Gad2-Fgf13 cKO interneurons (left). I-V curve for K$^+$ currents (right), (two-way ANOVA, *, p<0.05, WT N=3, n=26; KO N=5, n=29).

The online version of this article includes the following figure supplement(s) for figure 5:

**Figure supplement 1.** Inhibitory neuronal knockout of *Fgf13* does not result in sodium current deficits.

## Virally mediated re-expression of *Fgf13-S* and *Fgf13-VY* restores output to Gad2-Fgf13 cKO neurons

Since inhibitory neurons express both *Fgf13-S* and *Fgf13-VY*, we tested if re-expression of either of these isoforms in Gad2-Fgf13 cKO neurons could rescue the observed deficit in the input/output relationship. We isolated primary hippocampal neurons from Gad2-Fgf13 cKO or controls (*Gad2-Cre*) and infected them with the Cre-dependent AAV8-DIO-GFP virus along with either an AAV8-DIO-*Fgf13-S* or AAV8-DIO-*Fgf13-VY*. Immunocytochemistry of GFP$^+$ neurons showed expression of endogenous FGF13 in *Gad2-Cre* cultures and the absence of FGF13 in Gad2-Fgf13 cKO cultures (*Figure 6A*). Infection with *Fgf13-S* in a Gad2-Fgf13 cKO neuron showed concentrated expression in the axon initial segment and membrane-enriched signal in the soma and the dendritic branches (*Figure 6A*) while infection with *Fgf13-VY* produced prominent signal throughout the soma and branches, demonstrating successful re-expression of these splice variants in the knockout background. After viral re-expression of *Fgf13-VY* or *Fgf13-S*, neither threshold potential nor rheobase were affected and were not different compared to values from either WT or Gad2-Fgf13 cKO neurons (*Figure 6B*). After rescue with either *Fgf13-VY* or *Fgf13-S*, current injection >100 pA elicited more action potentials than in Gad2-Fgf13 cKO neurons, suggesting that excitability was at least partially restored (*Figure 6C*). Indeed, compared to Gad2-Fgf13 cKO neurons a smaller fraction of *Gad2*$^+$ cells re-expressing *Fgf13-VY* or *Fgf13-S* reached depolarization block when injected with >100 pA (*Figure 6D*), although neither *Fgf13-VY* nor *Fgf13-S* alone reduced the probability of conduction block to what was observed in WT neurons. Further analysis showed that, even though *Fgf13-S* or *Fgf13-VY* partially restored spiking, the action potential wave forms and the phase plots were not restored by either *Fgf13-S* or *Fgf13-VY* alone (*Figure 6E*). Examination of specific parameters revealed that neither *Fgf13-S* nor *Fgf13-VY* rescue was able to prevent the prolongation of APD50 nor reduction in the action potential amplitude, dV/dt max or action potential peak amplitude observed in Gad2-Fgf13 cKO neurons (*Figure 6F*), again suggesting that neither FGF13-S nor FGF13-VY is sufficient for maintaining excitability in *Gad2* neurons. On the other hand, infection with *Fgf13-S* partially hyperpolarized the interspike membrane potential (compared to Gad2-Fgf13 cKO, *Figure 6G*), suggesting that FGF13-S increased the repolarizing K$^+$ currents. Indeed, infection with *Fgf13-S* but not *Fgf13-VY* increased K$^+$ current density to WT levels (*Figure 6H*). These data suggest that FGF13-S and FGF13-VY, both expressed in inhibitory interneurons (*Joglekar et al., 2021*) contribute to inhibitory interneuron function and modulation of pyramidal neuron excitability.

## Discussion

While variants in or disruption of *FGF13* have been associated with DEEs (*Puranam et al., 2015*; *Fry et al., 2021*, *Velíšková et al., 2021*; *Narayanan et al., 2022*), the pathogenesis was unknown. Employing multiple cell type specific mouse knockout models complemented by isoform specific rescue strategies, our data reveal how *FGF13* ablation contributes to seizure susceptibility and provide new insights into FGF13 functions in neurons. We found that *Fgf13* ablation and consequent loss of FGF13-S and FGF13-VY in a subset of interneurons reduced overall inhibitory drive from *Fgf13*-expressing interneurons onto hippocampal pyramidal neurons. Rescue experiments suggest that both FGF13-S and FGF13-VY isoforms are required for proper interneuron function as re-expression of FGF13-S or FGF13-VY only partially restored excitability (our experimental setup did not allow rescue with both isoforms simultaneously). Building upon our recent analysis of *Fgf13* splice variant expression (*Joglekar et al., 2021*), our data resolve apparently conflicting reports in which reduced inhibitory interneuron function, suggested as the main contributor in a global *Fgf13* female heterozygous knockout model, (*Puranam et al., 2015*) contrasts with a model that proposes that *FGF13* variants

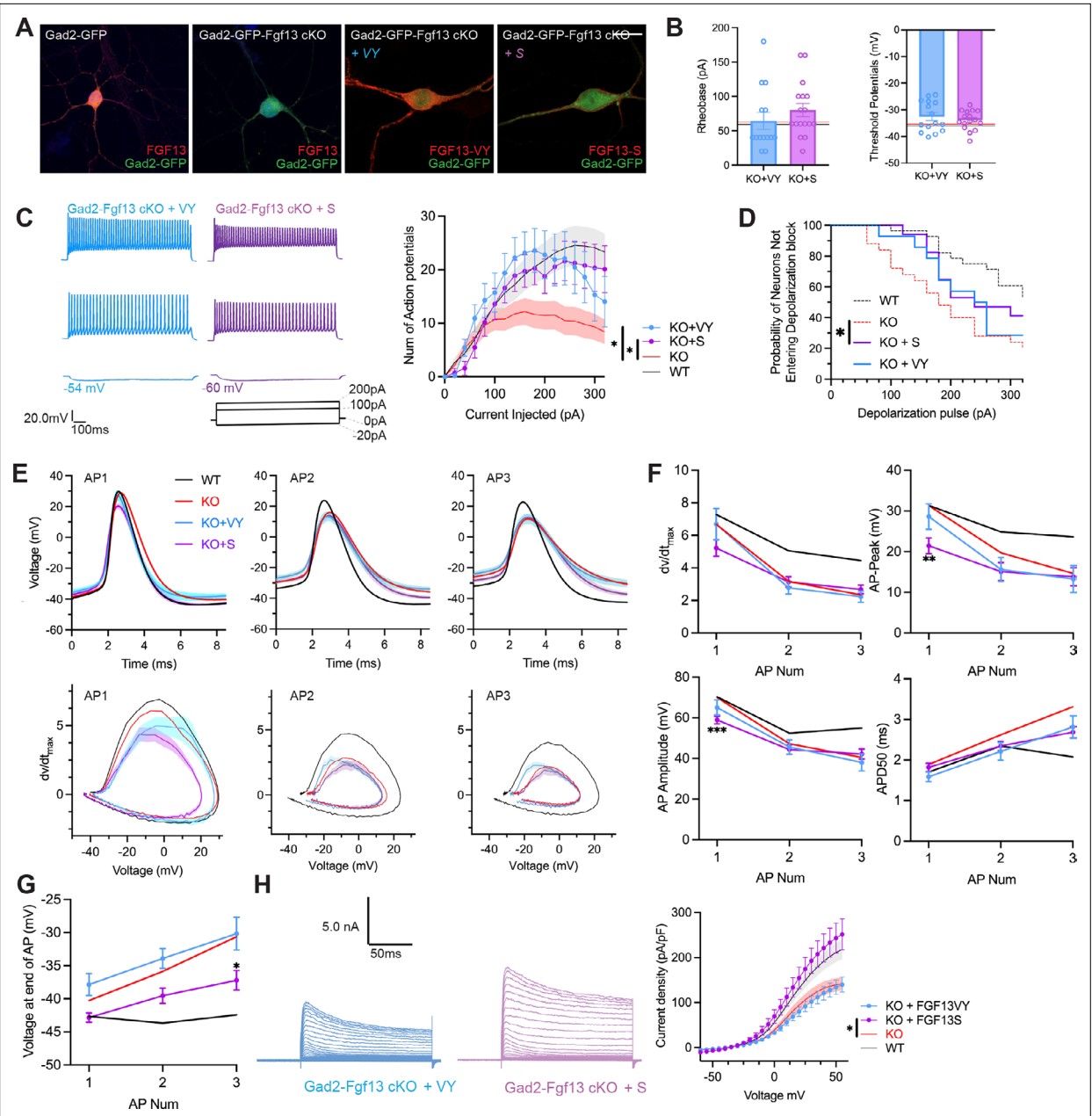

**Figure 6.** AAV-mediated expression of FGF13 isoforms rescues excitability deficits in Gad2-Fgf13 cKO neurons. (**A**) Examples of FGF13-stained neurons from primary hippocampal neuron cultures generated from Gad2-Fgf13 cKO male mice transduced with AAV8-DIO-GFP only, AAV8-DIO-GFP and AAV8-DIO-Fgf13-S, or AAV8-DIO-GFP and AAV8-DIO-Fgf13-VY (scale bar, 20 μm). (**B**) Gad2-Fgf13 cKO neurons expressing FGF13-VY or FGF13-S were not different in terms of threshold potential or rheobase, and were not different from wildtype (black line, from *Figure 5*) and Gad2-Fgf13 cKO neurons (red line, from *Figure 5*) (t-test, p=ns, KO +VY N=3, n=14; KO +S N=3 n=17). (**C**) Evoked action potential traces from Gad2-Fgf13 cKO interneurons expressing FGF13-VY or FGF13-S. Input-output curve shows increased firing of evoked action potentials from the FGF13-VY or FGF13-S expressing interneurons, relative to Gad2-Fgf13 cKO interneurons (red line, from *Figure 5*; black line = wild type, from *Figure 5*) (two-way ANOVA, *, p<0.05). (**D**) Gad2-Fgf13 cKO interneurons expressing FGF13-VY and FGF13-S do not enter depolarization block as early as Gad2-Fgf13 cKO interneurons (Red line [Gad2-Fgf13 cKO] and black line [wild type] are from *Figure 5*) (log-rank test, *, p<0.05). (**E**) Action potential wave forms and phase plots for the initial three action potentials of the spike train for FGF13-VY and FGF13-S rescued Gad2-Fgf13 cKO interneurons. The black and red lines are from *Figure 5*. (**F**) For the first three action potentials in the spike train, Gad2-Fgf13 cKO neurons re-expressed with FGF13-VY show no difference from Gad2-Fgf13 cKO neurons in terms of dV/dt max, AP peak, and AP amplitude, and AP50. Gad2-Fgf13 cKO neurons re-expressed with FGF13-S show difference from Gad2-Fgf13 cKO neurons only for the first action potential for AP peak and AP amplitude, but not dV/dt max or AP50. (two-way ANOVA, ***, p<0.001, KO +S vs. KO; **, p<0.01, KO +S vs. KO). (**G**) FGF13-S rescued neurons show a significant decrease in membrane voltage from Gad2-Fgf13 cKO neurons by the third action potential in the spike train (two-way ANOVA, *, p<0.05, KO +S vs. KO). (**H**) Example traces of K⁺ currents from Gad2-Fgf13 cKO

*Figure 6 continued on next page*

*Figure 6 continued*

neurons expressing FGF13-VY and FGF13-S. K$^+$ currents are rescued by expression of FGF13-S in Gad2-Fgf13 cKO interneurons (two-way ANOVA, *, p<0.05, KO +VY N=5, n=21, KO +S N=5, n=21).

within excitatory neurons drive increased excitability because of diminished long-term inactivation (*Fry et al., 2021*).

The apparent requirement of both FGF13-S and FGF13-VY extends previous studies showing their distinct roles in neurons (*Pablo et al., 2016*) and emphasizes that alternative splicing is an important source of protein diversity in the mammalian brain, an evolutionarily conserved feature thought to contribute to the nervous system's complexity (*Raj and Blencowe, 2015*; *Zhang et al., 2016*; *Karlsson and Linnarsson, 2017*; *Porter et al., 2018*; *Furlanis et al., 2019*; *Joglekar et al., 2023*). Cell-specific alternative splicing in the brain has been posited as a mechanism to efficiently adapt to the external environment by quickly modifying protein activity (*Lipscombe and Lopez Soto, 2019*). Not only does *FGF13* have the most extensive set of alternatively spliced isoforms among FHFs (*Smallwood et al., 1996*; *Munoz-Sanjuan et al., 2000*; *Pablo et al., 2016*), but long-range single-cell sequencing data show that *Fgf13* has a uniquely (among FHFs) segregated alternative splice variant expression pattern among distinct neuron cell type. Further, among all genes, *Fgf13* has one of the highest differential isoform expression (DIE) values across multiple neuronal cell-types (*Joglekar et al., 2021*).

Our results are also consistent with the available patient data. The responsible splice variant(s) were proposed to be either *FGF13-V*, *FGF13-VY*, and *FGF13-Y* that were lost in two affected brothers who inherited a maternal balanced translocation that disrupts *FGF13* on the X-chromosome and preserves the *FGF13-S* and the *FGF13-U* splice variants (*Puranam et al., 2015*) or, in contrast, *FGF13-S* in a series of unrelated individuals presenting with DEE symptoms (*Fry et al., 2021*). As our data suggest that both *FGF13-S* and *FGF13-VY* contribute within inhibitory interneurons, we provide a unifying explanation for those reports. Here, we were able to assess FGF13-S and FGF13-VY, chosen because they are most abundantly expressed isoforms in the adult mouse brain, but the inability to rescue electrophysiological consequences completely with either isoform alone leaves open the possibility that other isoforms (e.g. FGF13-U, FGF13-V) also make critical contributions.

These results also expand our understanding of how *FGF13* contributes to neuronal functions, which have been challenging to define because of *FGF13*'s location on the X chromosome, the embryonic lethality of complete knockout models, absence of cell-type-specific knockout models, *FGF13*'s complex alternative splicing, and an incomplete cataloging of specific functions for the various isoforms generated. Various roles for *FGF13* in inhibitory interneurons have previously been suggested. *Fgf13* silencing in *Nkx2.1-CreER* targeted neurons reduced the number of chandelier cell boutons (*Favuzzi et al., 2019*) and a more general role for *FGF13* in interneuron development was suggested by a human scRNA-seq dataset (*Herring et al., 2022*). Our data here, in which seizure susceptibility was attributable solely to *Fgf13* knockout in inhibitory interneurons and our demonstration that *Fgf13* knockout restricted to *Nkx2.1*-expressing neurons increased seizure susceptibility—albeit to a lesser extent than complete knockout in inhibitory interneurons—support not only a possible role for *Fgf13* in chandelier cell development but also confirm more general inhibitory interneuron roles. Along with the reduced seizure susceptibility of female heterozygote knockout animals, these data also imply a gene dosage effect within interneurons. The differences in survival among Gad2-Fgf13 cKO, Gad2-Fgf13 Het, Nkx2.1-Fgf13 cKO, and Nkx2.1-Fgf13 Het also suggest a gene-dosage effect. Although we were unable to generate female homozygous Gad2-Fgf13 cKO mice since the male hemizygous knockouts did not survive to adulthood, the reduced survival in male knockouts compared to female heterozygous knockouts further supports our hypothesis. The relative comprehensiveness of the specific Cre drivers also provides support for our hypothesis. In the hyperthermic seizure assay, mice from both the Nestin-Fgf13 cKO and Gad2-Fgf13 cKO lines, in which *Fgf13* was ablated in the entire interneuron population, suffered seizures at baseline (near 36 °C) as well as at elevated core body temperatures. In contrast, the majority of the Nkx2.1-Fgf13 cKO mice, generated with a more restrictive Cre driver, mostly suffered seizures only at elevated core body temperatures. We also observed occasional seizures in the home cage for all genotypes in which *Fgf13* was ablated in interneurons (Nestin-Fgf13 cKO, Nkx2.1-Fgf13 cKO, and Gad2-Fgf13 cKO) and often discovered dead mice, which we suspect was a SUDEP phenotype due to recurrent seizures, given that recurrent uncontrolled seizures are a major risk factor for sudden death (*Devinsky et al.,*

*2016*; *Richerson, 2023*). Indeed, long-term recording of Gad2-Fgf13 cKO mice (data not shown) confirmed that multiple spontaneous seizures occur in the knockouts prior to death in the home cage.

While our data provide a clear connection between ablation of *Fgf13* in inhibitory neurons and seizure susceptibility, we find that *Fgf13* ablation in excitatory neurons does not increase seizure susceptibility. Indeed, complete neuronal knockout with *Nestin-Cre* conferred a relative increased survival compared to the interneuron-specific knockout with *Gad2-Cre*. We speculate that this relative increased survival in the Nestin-Fgf13 cKO mice could be technical, due to less efficient targeting of interneurons by *Nestin-Cre* (*Liang et al., 2012*), but it may also be biological if *Nestin-Cre* targeting of both excitatory and inhibitory neurons is comparatively neuroprotective because of simultaneous loss of both excitation and inhibition and consequent compensation for the loss of inhibition. The previous observation that a global *Fgf13* heterozygous mouse line survived with a normal life span (*Puranam et al., 2015*) could support the interpretation that deletion from other cell types is comparatively neuroprotective. Moreover, our observations that our Emx1-Fgf13 cKO mice do not suffer seizures is consistent with an earlier study of Emx1-Fgf13 cKO mice, in which seizures were not reported (*Wu et al., 2012*). Previous studies suggested that FGF13-S, the isoform most highly expressed in excitatory neurons, confers long-term inactivation (*Venkatesan et al., 2014*; *Fry et al., 2021*), and the reduction of long-term inactivation due to DEE-associated variants drives increased excitability as the mechanism for seizures (*Fry et al., 2021*). Our data confirm the ability of FGF13-S to impart long-term inactivation and that ablation of *Fgf13* in excitatory neurons eliminates long-term inactivation, yet this mechanism does not likely contribute to seizure susceptibility. Rather, knockout of *Fgf13* in central nervous system excitatory neurons appears to affect learning and memory (*Wu et al., 2012*), which we did not assess here. An important limitation of our knockout models, however, is that they cannot provide direct insight into how missense variants reported in patients contribute to seizures. Nevertheless, since our Emx1-Fgf13 cKO mice show a loss of long-term inactivation in excitatory neurons—the proposed mechanism by which the missense variants drive seizures—our model provides new boundaries on how those missense variants act in excitatory neurons.

Further, other than long-term inactivation, data herein show that *Fgf13* ablation led to relatively minor effects on neuronal voltage-gated Na$^+$ currents in *Emx1$^+$* excitatory neurons compared to previous results from others and us (*Venkatesan et al., 2014 Barbosa et al., 2017*; *Pablo et al., 2016*; *Wang et al., 2021*; *Shen et al., 2022*). Those neuronal studies investigated DRG neurons, or CNS neurons via RNAi-mediated knockdown of infusion of antibodies targeting the FGF13-S N-terminus, and not a cell-type-specific knockout as we studied here, thus suggesting differences in the models studied. As the *FGF13* DEEs result from a stable germ-line variants, the knockout model here (in contrast to knockdown approaches in cultured CNS neurons or acute infusion of antibodies) likely represents a more relevant disease model.

How does loss of FGF13 in inhibitory interneurons confer seizure susceptibility? Although knockout of *Fgf13* in *Gad2$^+$* neurons decreased excitability and reduced overall inhibitory drive to pyramidal cells, knockout surprisingly did not affect Na$_V$ currents in inhibitory neurons. This conclusion fits well with our previous report that the cytoplasmic C-terminal in Na$_V$1.1, the major Na$_V$ channel expressed in inhibitory neurons, has a substitution of an amino acid that reduces FHF binding (*Wang et al., 2012*), thus rendering these Na$_V$1.1 channels—and thus Na$_V$ current in inhibitory interneurons—poor targets for FHF regulation. Rather, we found that perturbed excitability in interneurons lacking *Fgf13* results from reduced K$^+$ currents. This fits with our demonstration in cardiomyocytes that loss of FGF13 similarly reduces repolarizing K$_V$ currents (*Wang et al., 2017*). Moreover, these data build upon the growing recognition that FHFs exert effects beyond Na$_V$ channel regulation—the focus of most FHF studies since FHFs were identified as capable of directly interacting with Na$_V$ channel C-termini (*Liu et al., 2001*). Other identified roles for FGF13, such as regulating microtubule stability (*Wu et al., 2012*), affecting ribosome biogenesis (*Bublik et al., 2017*), or generating axoaxonic synapses in chandelier cells (*Favuzzi et al., 2019*) may also contribute, but we did not investigate those roles here. The mechanism(s) by which FGF13 affects K$_V$ currents is not known—we have not detected direct interaction between FHFs and K$_V$ channels—but we speculate that the identified role for FGF13 in stabilizing microtubules may contribute, as microtubules regulate trafficking of ion channels to the plasma membrane during channel biogenesis (*Vacher et al., 2008*).

Although we did not test it directly, our data are consistent with a possible role for FGF13 in chandelier cell function. Chandelier cells are fast-spiking GABAergic interneurons that uniquely target the

axon initial segment of pyramidal cells. Chandelier cells are found sparsely in the brain, but are highly branched, enabling contact on hundreds of neighboring pyramidal cells (*Compans and Burrone, 2023*). *Fgf13* has previously been identified as a key molecule in the development of chandelier cell axons (*Favuzzi et al., 2019*), but whether *Fgf13* subsequently contributes to their function has not been reported. Chandelier cells originate from the MGE, alongside parvalbumin- and somatostatin-expressing interneurons (*Gallo et al., 2020*). As our Nkx2.1-Fgf13 cKO mice target deletion of *Fgf13* from all the above subtypes, our data suggest that FGF13 may regulate chandelier cell function in seizure generation. Although chandelier cells have been shown to depolarize pyramidal cells and therefore exert a pro-excitatory role in the cortex (*Szabadics et al., 2006*), we observe otherwise in our mouse model in which decreased interneuron excitability results in a pro-excitatory state.

Understanding the molecular mechanisms that drive the pathogenesis of *Fgf13*-related seizures will enhance our understanding of DEEs and epilepsy at large. Future work will reveal the mechanisms of FGF13 interaction with potassium channels or other molecules that enable interneuron repolarization and excitability.

## Methods
### Animals
Mice were handled in accordance with the ethical guidelines of the National Institutes of Health Guide for the Care and Use of Laboratory Animals. This study was approved by the Weill Cornell Medical Center Institutional Animal Care and Use Committee (Protocol 2016–0042). Genetically modified mice were maintained on C57BL6/J background (000664; The Jackson Laboratory, Bar Harbor, ME, USA). All mice were maintained on a standard rodent chow diet (PicoLab Rodent Diet 20; 5053; LabDiet, St. Louis, MO, USA) with a 12 hr light/dark cycle. Mice were weaned at post-natal day 21 (P21) and group housed in cages holding between two and five mice. To generate neuron-specific mutant mice, female *Fgf13*$^{fl/fl}$ mice (*Wang et al., 2017*) were crossed with male *Nestin-Cre*$^{+/-}$ (*B6.Cg-Tg(Nes-cre)1Kln/J*, JAX #003771) to produce male hemizygous knockouts (Nestin-Fgf13 cKO), female heterozygous mutants (Nestin-Fgf13 Het), and wild type littermates. To generate excitatory neuron-targeted mutant mice, female *Fgf13*$^{fl/fl}$ mice were crossed with male *Emx1-Cre*$^{+/-}$ (*B6.129S2-Emx1tm1(cre)Krj/J*, #005628) to produce male hemizygous knockouts (Emx1-Fgf13 cKO), female heterozygous mutants (Emx1-Fgf13 Het), and wild type littermates. Female *Emx1* knockouts were generated by crossing male hemizygous knockouts (Emx1-Fgf13 cKO) with female *Fgf13*$^{fl/fl}$ mice. To generate interneuron-specific mutant mice, male *Gad2-Cre*$^{+/-}$ (*Gad2tm2(cre)Zjh/J*, #010802) were crossed with female *Fgf13*$^{fl/fl}$ mice to produce male hemizygous knockouts (Gad2-Fgf13 cKO), female heterozygous mutants (Gad2-Fgf13 Het), and wild type littermates. To generate interneuron-GFP reporter mice, male *Gad2-Cre*$^{+/+}$ (*Gad2tm2(cre) Zjh/J*, #010802) were crossed with *Cre*-dependent Ai6$^{+/+}$ mice (B6.Cg-Gt(ROSA)26Sortm6(CAG-ZsGreen1)Hze/J), and the resulting Gad2-GFP (*Gad2-Cre*$^{+/-}$; Ai6$^{+/-}$) males were crossed with female *Fgf13*$^{fl/fl}$ mice to generate *Fgf13* knockout interneuron-GFP reporter mice (*Gad2-Cre*$^{+/-}$; *Fgf13*$^{FL/Y}$; Ai6$^{+/-}$-or Gad2-GFP-Fgf13 cKO). To generate medial ganglionic eminence (MGE) derived interneuron-targeted mutant mice, female *Fgf13*$^{fl/fl}$ mice were crossed with male *Nkx2.1-Cre*$^{+/-}$ (*C57BL/6J-Tg(Nkx2-1-cre)2Sand/J*, # 008661) to produce male hemizygous knockouts (Nkx2.1-Fgf13 cKO), female heterozygous mutants (Nkx2.1-Fgf13 Het), and wild type littermates.

### Hyperthermia-induced seizures
Mice at P12 were tested using a modified protocol (*Cheah et al., 2012*; *Puranam et al., 2015*). Mouse core body temperature was recorded with a rodent rectal temperature probe placed at a depth of 1 cm and connected to a rodent temperature controller (TCAT-2DF, Physitemp). Behavior was monitored by a video camera (Logitech). Command temperature ±0.3 °C was maintained using an infrared heat lamp positioned directly over the recording chamber (2 L glass beaker). Mice were acclimated to the chamber for 5 minutes, and any mice with a lower core body temperature were warmed with the heat lamp up to 36.5 °C. For baseline monitoring, temperature was adjusted to an initial set-point of 36.5 °C for 5 min. Body temperature was elevated in 0.5 °C increments, for 2 min at each increment. Temperature was increased until a seizure or 42.0 °C was reached. At the end of hyperthermia induction, mice were placed on a room temperature surface to recover and monitored for 10 min.

## Electroencephalogram recording

### Animal preparation

Mice (P12) were anesthetized under isoflurane (3.5% induction, 1–2% maintenance), provided pre-surgical analgesia (Meloxicam, 2 mg/kg SC; Bupivicaine 0.15 ml under scalp), and head-fixed in a stereotaxic frame (KOPF Instruments) with ear bars and an anesthesia-passing nose cone. A heatpad was placed under the animal and the eyes were protected with ointment (Paralube Vet). Hair was removed with fine scissors and an oval-shaped incision (~4 × 6 mm) was made on the scalp to expose the skull. Fine forceps were used to scrape the membrane, remove any remaining hair, and expose bregma and skull sutures. The hippocampal EEG site was located at –2.5 mm AP, 1.5 mm ML, –1.25 mm DV. A dental drill was then used to open small craniotomies at the (AP, ML) sites for EEG contacts with a 0.5 mm bur (Meisinger). Reference and ground contact craniotomies were made on the contralateral skull. The EEG recording system (Pinnacle Technology Inc) was attached to headmounts (Pinnacle product #8201-ss) with conductive metallic holes and fastened to the skull by small conductive screws (Pinnacle product #8209). An insulated tungsten wire (0.002″, California Fine Wire Company) was soldered to the EEG mount and trimmed to 2 mm to target the hippocampus (*Klorig et al., 2019*), using epoxy to insulate the soldered portion of the electrode. The mount, with hippocampus electrode attached, was slowly lowered onto the skull with a micromanipulator such that the electrode was inserted directly down into the craniotomy over hippocampus. The headmount was then attached with a small amount of dental cement (C&B Metabond, Parkell) to hold the device steady as skull screws were then implanted into the remaining craniotomies. Skull screws were then reinforced to the headmount by conductive silver paint (SPI supplies). Dental cement (Metabond) was then used to cover all parts of the device and attach it firmly to the skull.

### Recording and analysis

After recovery from surgery the headmount was attached to a preamplifier and animals placed in a bedded recording chamber with ad libitum food and water. Pinnacle EEG software recorded electrographic data at 400 Hz from the tethered preamplifier. Spontaneous activity was recorded for up to 5 hr, while heat-lamp experiments were recorded as described above. After recording, data were transferred to a PC and analyzed in MATLAB (Mathworks). Raw data were bandpass filtered (1–60 Hz, infinite impulse response filter with bandpass.m) and events of interest were automatically flagged by using line length of the signal (difference of successive voltage values with a moving 5 s window), which permitted detection of large deviations (MATLAB findpeaks.m). Interictal and seizure events were then visually examined. In heat-lamp experiments, the times of temperature steps were recorded for alignment with the simultaneous EEG recording.

## Hippocampal neuronal cultures

Hippocampi were dissected from P0-1 newborn pups and dissociated through enzymatic treatment with 0.25% trypsin and subsequent trituration. The cells were plated on glass coverslips previously coated with poly-D-lysine and laminin in 24-well cell culture plates. The hippocampal cells were grown in neurobasal A medium (ThermoFisher Scientific) supplemented with 2% B-27, 2 mM glutamine, 10% heat-inactivated fetal bovine serum, and 1% penicillin/streptomycin in a 5% $CO_2$ incubator at 37 °C overnight. After 24 hr, this medium was replaced by a culture medium containing 2% B-27, 0.5 mM glutamine, 1% heat-inactivated fetal bovine serum, 70 μm uridine and 25 μm 5-fluorodeoxyuridine. Cultured neurons were used for electrophysiology and immunocytochemistry.

## Immunocytochemistry

Cells were fixed in 4% PFA 20 min, rinsed three times with phosphate buffered saline (PBS), and blocked in 2.5% bovine serum albumin with 0.2% Triton-X in PBS for 30 min. Primary anti-FGF13 antibody (*Wang et al., 2011*) was diluted in 2.5% bovine serum albumin and incubated in 4 °C overnight. Coverslips were rinsed three times in PBS, then incubated in Alexa-568 secondary antibody (Thermo Fisher Scientific Cat# A10042, RRID:AB_2534017 1:500 in 2.5% bovine serum albumin) for 1 hur at room temperature followed by DAPI for 5 min. Coverslips were mounted onto glass slides with mounting media (Vector Laboratories).

## Acute slice preparation

Acute hippocampal slices were prepared from P12 to P14 mice. Briefly, after the mouse was anesthetized with isoflurane and then decapitated, the brain was quickly extracted and transferred into ice-cold cutting solution bubbled with 95% $O_2$ and 5% $CO_2$. The cutting solution contained (in mM): NaCl 77, sucrose 75, KCl 2.5, $NaH_2PO_4$ 1.4, $NaHCO_3$ 25, $MgSO_4$ 7, $CaCl_2$ 0.5, glucose 25, and sodium pyruvate 3, pH 7.4. Coronal brain slices (300 μm) including hippocampus were prepared using a Leica VT 1200 S vibratome (Leica, Inc), and were incubated in a BSK-2 brain slice keeper (Automate Scientific, CA) containing oxygenated artificial cerebrospinal fluid (aCSF) at 35 °C for 30 min to recover. Afterwards, the slices were maintained at room temperature at least 30 min before use.

## Electrophysiology

For spontaneous excitatory and inhibitory synaptic currents (sEPSCs and sIPSCs) in brain slices, the slice was placed in a recording chamber on the stage of an upright, infrared-differential interference contrast microscope (BX51WI, Olympus Optical) equipped with an ORCA-Flash2.8 C11440 Digital CMOS Camera (Hamamatsu Photonics), and was continuously perfused at a rate of 2 ml/min with aCSF bubbled with 95% O2 and 5% CO2 at 35 °C. Pyramidal neurons in hippocampal CA1 stratum pyramidale were visualized with a 40 X water-immersion objective. Spontaneous EPSCs and IPSCs were recorded in the whole-cell voltage clamp configuration at a holding potential of –70 mV and sampled at 10 kHz and filtered at 2 kHz using an Axopatch 200B amplifier and Digidata 1322 A digitizer (Molecular Devices). The pipette internal solution for sEPSC contained (in mM): potassium gluconate (KGlu) 125, KCl 10, $MgCl_2$ 5, EGTA 0.6, HEPES 5, $CaCl_2$ 0.06, phosphocreatine disodium 10, Mg-ATP 2, $Na_2$-GTP 0.2, creatine phosphokinase 50 U/ml, and N-Ethyl lidocaine bromide 5, osmolarity 289 mOsm/L, pH 7.2 adjusted with KOH. The pipette internal solution for sIPSC contained (in mM): KCl 125, $MgCl_2$ 5, EGTA 0.6, HEPES 5, $CaCl_2$ 0.06, phosphocreatine disodium 10, Mg-ATP 2, $Na_2$-GTP 0.2, creatine phosphokinase 50 U/ml and N-Ethyl lidocaine bromide 5, osmolarity 291 mOsm/L, pH 7.2 adjusted with KOH. The external solution aCSF contained (in mM): NaCl 126, KCl 2.5, $NaH_2PO_4$ 1.25, $NaHCO_3$ 26, $CaCl_2$ 2, $MgSO_4$ 2, glucose 10. Bicuculline methiodide 20 μM was added to aCSF for sEPSC, and 2-amino-5-phosphonovaleric acid (APV) 50 μM and 6,7-dinitroquinoxaline-2,3-dione (DNQX) 20 μM was added to aCSF for sIPSC.

Whole-cell sodium $Na^+$ currents and whole-cell $K^+$ currents, or action potentials (APs) from cultured neurons were recorded with a HEKA EPC10 amplifier in the voltage-clamp or current-clamp configuration, respectively.

For $Na^+$ current recordings in cultured neurons, the internal pipette solution contained (in mM): CsF 125, NaCl 10, HEPES 10, TEA-Cl 15, EGTA 1.1, Na-GTP 0.5, pH 7.4 with NaOH; the external solution contained NaCl 125, KCl 5, $CaCl_2$ 2, $MgCl_2$ 1, TEA-Cl 20, HEPES 5, Glucose 10, pH 7.4 with NaOH. For K+current recordings and AP initiation in cultured neurons, the internal pipette solution contained (in mM): potassium gluconate 130, KCl 10, $MgCl_2$ 5, ethylene glycol-bis(β-aminoethyl ether)-*N,N,N',N'*-tetraacetic acid 0.6, HEPES 5, $CaCl_2$ 0.06, phosphocreatine disodium 10, Mg-ATP 2, $Na_2$-GTP 0.2, and creatine phosphokinase 50 U/ml, pH 7.2 adjusted with KOH; the external solution contained (in mM): NaCl 119, KCl 5, HEPES 20, $CaCl_2$ 2, $MgCl_2$ 2, glucose 30, APV 0.05, DNQX 0.02, bicuculline 0.02, pH 7.3 adjusted with NaOH. Additionally for K+current recordings, the external solution contained 100 nM tetrodotoxin (TTX) and 5 μM nifedipine to block $Na^+$ and $Ca^{2+}$ channels.

Recording pipettes were pulled from borosilicate glass with Sutter P-97 Micropipette Puller (Sutter Instrument Co.). For $Na^+$ and $K^+$ current recordings, pipette resistance ranged from 1.5 to 2.5 MΩ and the series resistance was compensated by at least 70%. For AP initiation recordings, pipette resistance ranged from 2.0 to 3.0 MΩ. For recordings of sEPSCs and sIPSCs in brain slices, pipette resistance ranged from 2.6 to 6.1 MΩ and series resistance was 10.5±0.5 MΩ for *WT* and 10.8±0.4 MΩ for Gad2-Fgf13 cKO without compensation.

## Analysis of electrophysiological data

Action potentials were elicited by holding the cell at 0 pA in current clamp mode and then stimulating with current increments for 1 s. The numbers of action potentials elicited for each current injection were quantified using Fitmaster software. Action potential parameters were analyzed using Clampfit 10.4 software. Specifically, action potential amplitude and action potential duration measured at 50% of the amplitude (APD50) values for the first three action potentials were determined using

the statistics function. Additionally, dV/dt values were computed using the differential function. The threshold potential was determined by analyzing the phase plot of the initial action potential at rheobase by identifying the voltage value on the x-axis where the y-axis reached a rate of 1 mV/ms.

Na$^+$ and K$^+$ channel currents were activated by holding the neuron at −90 mV and applying incremental voltages for a duration of 150ms. Fitmaster software was used for the quantification of various channel parameters. The non-inactivating portion of the K$^+$ channel current was assessed by analyzing the peak current at the conclusion of 150ms depolarizing pulses. To generate steady-state inactivation curves, currents were elicited at a holding potential of −120 mV followed by a 500 ms prepulse to voltages ranging from −120 to +20 mV. Subsequently, currents were measured at −20 mV for 20ms. Normalized current values obtained at −20 mV were utilized to construct the steady-state inactivation curves. A Boltzmann equation, $[I/Imax = (1+\exp((V − V1/2) / k))−1]$, was employed to fit the data, where Imax denotes the maximum current, V1/2 represents the half-inactivation voltage, and k signifies the slope. This equation facilitated the calculation of V1/2 of inactivation.

Data for Na$^+$ and K$^+$ current or APs were analyzed with Axon Clampfit (Molecular Devices), and data for sEPSC and sIPSC were analyzed with Mini Analysis (Synaptosoft).

## Immunohistochemistry

Mice were sacrificed at postnatal day 13–16 (P13-16). Brains were extracted after transcardial perfusion of PBS followed by 4% paraformaldehyde (PFA). They were incubated overnight rocking at 4 °C in 4% paraformaldehyde, then equilibrated in 15% sucrose followed by 30% sucrose. Brains were embedded in Optimal Cutting Temperature (OCT) compound in cryomolds and cryosectioned at 50 µm into 0.1 M PBS. Floating sections were washed three times with PBS, then blocked for 1 hr in 2.5% bovine serum albumin and 0.2% Triton-X in PBS. Tissue was incubated with the anti-FGF13 antibody overnight at 4 °C. After sections were washed three times with PBS, they were incubated with Alexa-568 secondary (Thermo Fisher Scientific Cat# A10042, RRID:AB_2534017 1:500 in 2.5% bovine serum albumin) antibody for 2 hr at room temperature. Sections mounted onto glass slides with mounting media (Vector laboratories) and sealed with glass cover slips. Slides were stored at 4 °C. Hippocampal tissue was imaged using a confocal microscope (Zeiss LSM 880) and analyzed using Image-J software.

## Western blot

Tissue was immersed in ice cold RIPA buffer supplemented with Halt protease and phosphatase inhibitor cocktails (Thermo Fisher Scientific) and placed in a manual Dounce tissue homogenizer, in which they were homogenized for 1 min on ice. The homogenate was then centrifuged at 21,000 x *g* at 4 °C. Supernatant was collected and protein concentration was determined by bicinchoninic acid assay. Protein was separated on an 8–16% gradient Tris-glycine gels and transferred to PVDF blotting membrane (GE Healthcare Life Sciences). The membrane was immunoblotted with 1:200 anti-FGF13 antibody or 1:200 anti-pan FHF-A (Neuromab clone N235/22) for 2 hr at RT to immunoblot for FGF13-S. Vinculin (Sigma V9131) was used as a loading control. The blots were visualized by chemiluminescence and images were captured using ChemiDoc Tough Imaging System (Bio-Rad).

## BaseScope in situ hybridization

Mice were sacrificed at P13-15. Brains were extracted after transcardial perfusion of PBS followed by 4% paraformaldehyde. The tissue was incubated for 24 hr rocking at 4 °C in 4% paraformaldehyde, then equilibrated in 15% sucrose followed by 30% sucrose. Brains were embedded in OCT compound in cryomolds and sectioned coronally in 14 µm sections onto SuperFrost (Thermo Fisher) slides, then stored at −80 °C up to 3 months. Custom *Fgf13* isoform probes were generated as previously described (*Joglekar et al., 2021*) and used according to the manufacturer's protocol for fixed frozen tissues. Hematoxylin followed by 0.02% ammonia water was used for nuclear counterstain.

## Targeted metabolomics

Mice were sacrificed at P13 and brains were immediately extracted, homogenized, and placed into pre-chilled 80% methanol (−80 °C). The extract was dried with a Speedvac, and redissolved in HPLC grade water before it was applied to the hydrophilic interaction chromatography LC-MS. Metabolites were measured on a Q Exactive Orbitrap mass spectrometer (Thermo Fisher Scientific), which was

coupled to a Vanquish UPLC system (Thermo Fisher Scientific) via an Ion Max ion source with a HESI II probe (Thermo Fisher Scientific). A Sequant ZIC-pHILIC column (2.1 mm i.d. ×150 mm, particle size of 5 μm, Millipore Sigma) was used for separation of metabolites. A 2.1 × 20 mm guard column with the same packing material was used for protection of the analytical column. Flow rate was set at 150 μL/min. Buffers consisted of 100% acetonitrile for mobile phase *A*, and 0.1% $NH_4OH$/20 mM $CH_3COONH_4$ in water for mobile phase *B*. The chromatographic gradient ran from 85% to 30% *A* in 20 min followed by a wash with 30% *A* and re-equilibration at 85% *A*. The Q Exactive was operated in full scan, polarity-switching mode with the following parameters: the spray voltage 3.0 kV, the heated capillary temperature 300 °C, the HESI probe temperature 350 °C, the sheath gas flow 40 units, the auxiliary gas flow 15 units. MS data acquisition was performed in the m/z range of 70–1000, with 70,000 resolution (at 200 m/z). The AGC target was 1e6 and the maximum injection time was 250ms. The MS data was processed using Xcalibur 4.1 (Thermo Fisher Scientific) to obtain the metabolite signal intensities. Identification required exact mass (within 5 ppm) and standard retention times. Statistical analysis was performed as follows: The intensity values were subjected to a Log2 transformation to normalize the data distribution. Student's *t*-test was conducted to calculate the significance of observed differences between groups. Enrichment analyses were performed using Metabolanalyst online tools (https://www.metaboanalyst.ca/).

## Reverse transcriptase quantitative polymerase chain reaction

From hippocampal neuron cultures prepared for electrophysiology, RNA was isolated using RNeasy Mini Kit (QIAGEN) according to the manufacture's instructions. Reverse transcription to generate cDNA was performed using SuperScript IV VILO Master Mix with ezDNase (Invitrogen). qPCR was performed in duplicate for each sample with QuantStudio 3 (Applied Biosystems) using SYBR green-based detection chemistries (Bio-Rad). For primer sequences used for qPCR target genes, see *Supplementary file 1*.

## Statistics

Statistical analyses were performed in GraphPad Prism v10 and specific analyses are described in the accompanying figure legends.

## Acknowledgements

Supported by T32 DA03980 and F31 DA053796 (SL) and R01 HL160089 (GSP). We thank Andrew S Lee and Gülcan Akgül for critical reading of the manuscript, as well as Mattia Malvezzi for thoughtful discussions.

## Additional information

### Competing interests

Geoffrey S Pitt: is a scientific advisory board member for Tevard Biosciences. The other authors declare that no competing interests exist.

### Funding

| Funder | Grant reference number | Author |
| --- | --- | --- |
| National Institute on Drug Abuse | T32 DA03980 | Susan Lin |
| National Institute on Drug Abuse | F31 DA053796 | Susan Lin |
| National Heart, Lung, and Blood Institute | R01 HL160089 | Geoffrey S Pitt |

The funders had no role in study design, data collection and interpretation, or the decision to submit the work for publication.

## Author contributions
Susan Lin, Formal analysis, Investigation, Writing – original draft, Writing – review and editing; Aravind R Gade, Formal analysis, Investigation, Writing – review and editing; Hong-Gang Wang, Patrick Towers, Maiko Matsui, Formal analysis, Investigation; James E Niemeyer, Allison Galante, Isabella DiStefano, Jorge Nunez, Investigation; Theodore H Schwartz, Anjali Rajadhyaksha, Supervision; Geoffrey S Pitt, Conceptualization, Formal analysis, Supervision, Writing – original draft, Project administration, Writing – review and editing

## Author ORCIDs
Susan Lin ⬤ https://orcid.org/0000-0001-9344-6594
Maiko Matsui ⬤ https://orcid.org/0000-0001-9706-8244
Geoffrey S Pitt ⬤ https://orcid.org/0000-0003-2246-0289

## Ethics
Mice were handled in accordance with the ethical guidelines of the National Institutes of Health Guide for the Care and Use of Laboratory Animals. This study was approved by the Weill Cornell Medical Center Institutional Animal Care and Use Committee (Protocol Number: 2016-0042).

Reviewer #2 (Public review): https://doi.org/10.7554/eLife.98661.3.sa1
Reviewer #3 (Public review): https://doi.org/10.7554/eLife.98661.3.sa2
Author response https://doi.org/10.7554/eLife.98661.3.sa3

---

# Additional files

## Supplementary files
Supplementary file 1. Primers used for real time quantitative polymerase chain reaction. Forward and reverse primer pair sequences and, where relevant, the PrimerBank ID.

Supplementary file 2. Metabolite set enrichment analysis in Gad2-Fgf13 cKO vs. WT. Unbiased metabolites identified by mass spectometery in Gad2-Fgf13 cKO vs. WT.

MDAR checklist

## Data availability
All data generated or analysed during this study are included in the manuscript and supporting files.

The following previously published datasets were used:

| Author(s) | Year | Dataset title | Dataset URL | Database and Identifier |
|---|---|---|---|---|
| Harris K, Hochgerner H, Skene NG, Magno L, Katona L, Bengtsson Gonzales C, Lonnerberg P, Kessaris N, Linnarsson S, Hjerling-Leffler J | 2018 | Transcriptomic analysis of CA1 inhibitory interneurons | https://doi.org/10.6084/m9.figshare.6198656.v1 | figshare, 10.6084/m9.figshare.6198656.v1 |
| Harris KD, Bengtsson Gonzales C, Hochgerner H, Skene NG, Magno L, Katona L, Somogyi P, Kessaris N, Linnarsson S, Hjerling-Leffler J | 2018 | Single-cell transcriptomic analysis of mouse CA1 inhibitory neurons | https://www.ncbi.nlm.nih.gov/geo/query/acc.cgi?acc=GSE99888 | NCBI Gene Expression Omnibus, GSE99888 |

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
