## [Editor Report · eLife Assessment]

This **important** study advances our understanding of how FGF13 variants confer seizure susceptibility. By acting in a set of inhibitory interneurons, FGF13 regulates synaptic transmission and excitability. The data presented here are **convincing** and combine cell type-specific knockouts and electrophysiology, complemented by histology/RNA studies. Collectively, this research will be of interest to a wide audience, particularly those involved in the study of epilepsy, inhibitory neurons, and ion channels.

---

## [Referee Report · Reviewer #2 (Public review)]

Summary

The authors address three primary questions:

(1) how FGF13 variants confer seizure susceptibility,

(2) the specific cell types involved, and

(3) the underlying mechanisms, particularly regarding Nav dysfunction.

They use different Cre drivers to generate cell type-specific knockouts (KOs). First, using Nestin-Cre to create a whole-brain Fgf13 KO, they observed spontaneous seizures and premature death. While KO of Fgf13 in excitatory neurons does not lead to spontaneous seizures, KO in inhibitory neurons recapitulates the seizures and premature death observed in the Nestin-Cre KO. They further narrow down the critical cell type to MGE-derived interneurons (INs), demonstrating that MGE-neuron-specific KO partially reproduces the observed phenotypes. "All interneuron" KOs exhibit deficits in synaptic transmission and interneuron excitability, not seen in excitatory neuron-specific KOs. Finally, they rescue the defects in the interneuron-specific KO by expressing specific Fgf13 isoforms. This is an elegant and important study adding to our knowledge of mechanisms that contribute to seizures.

Strengths

• The study provides much-needed cell type-specific KO models.

• The authors use appropriate Cre lines and characterize the phenotypes of the different KOs.

• The metabolomic analysis complements the rest of the data effectively.

• The study confirms and extends previous research using improved approaches (KO lines vs. in vitro KD or antibody infusion).

• The methods and analyses are robust and well-executed.

Weaknesses

• One weakness lies in the use of the Nkx2.1 line (instead of Nkx2.1CreER) in the paper. As a result, some answers to key questions are incomplete. For instance, it remains unclear whether the observed effects are due to Chandelier cells or NGFCs, potentially both MGE and CGE derived, explaining why Nkx2.1 alone does not fully replicate the overall inhibitory KO. Using Nkx2.1CreER could have helped address the cell specificity. With the Nkx2.1 line used in the paper, the answer is partial.

• While the mechanism behind the reduced inhibitory drive in the IN-specific KO is suggested to be presynaptic, the chosen method does not allow them to exactly identify the mechanisms (spontaneous vs mEPSC/mIPSC), and whether it is a loss of inhibitory synapses (potentially axo-axonic) or release probability.

General Assessment

The general conclusions of this paper are supported by data. As it is, the claim that "these results enhance our understanding of the molecular mechanisms that drive the pathogenesis of Fgf13-related seizures" is partially supported. A more cautious term may be more appropriate, as the study shows the mechanism is not Nav-mediated and suggests alternative mechanisms without unambiguously identifying them. The conclusion that the findings "expand our understanding of FGF13 functions in different neuron subsets" is supported, although somewhat overstated, as the work is not conclusive about the exact neuron subtypes. However, it does indeed show differential functions for specific neuronal classes, which is a significant result.

Impact and Utility

This paper is undoubtedly valuable. Understanding that excitatory neurons are not the primary contributors to the observed phenotypes is crucial. The finding that the effects are not MGE-unique is also important. This work provides a solid foundation for further research and will be a useful resource for future studies.

---

## [Referee Report · Reviewer #3 (Public review)]

Summary:

The authors aimed to determine the mechanism by which seizures emerge in Developmental and Epileptic Encephalopathies caused by variants in the gene FGF13. Loss of FGF13 in excitatory neurons had no effect on seizure phenotype as compared to loss of FGF13 in GABAergic interneurons, which in contrast caused a dramatic proseizure phenotype and early death in these animals. They were able to show that Fgf13 ablation and consequent loss of FGF13-S and FGF13-VY reduced overall inhibitory input from Fgf13-expressing interneurons onto hippocampal pyramidal neurons. This was shown to occur not via disruption to voltage gated sodium channels but rather by reducing potassium currents and action potential repolarisation in these interneurons.

Strengths:

The authors employed multiple well validated, novel mouse lines with FGF13 knocked out in specific cell types including all neurons, all excitatory cells, all GABAergic interneurons, or a subset of MGE-derived interneurons, including axo-axonic chandelier cells. The phenotypes of each of these four mouse lines were carefully characterised to reveal clear differences with the most fundamental being that Interneuron-targeted deletion of FGF13 led to perinatal mortality associated with extensive seizures and impaired the hippocampal inhibitory/excitatory balance while deletion of FGF13 in excitatory neurons caused no detectable seizures and no survival deficits.

The authors made excellent use of western blotting and in situ hybridisation of the different FGF13 isoforms to determine which isoforms are expressed in which cell types, with FGF3-S predominantly in excitatory neurons and FGF13-VY and FGF13-V predominantly in GABAergic neurons.

The authors performed highly detailed electrophysiological analysis of excitatory neurons and GABAergic interneurons with FGF13 deficits using whole-cell patch clamp. This enabled them to show that FGF13 removal did not affect voltage-gated sodium channels in interneurons, but rather reduced the action of potassium channels, with the resultant effect of making it more likely that interneurons enter depolarisation block. These findings were strengthened by the demonstration that viral re-expression of different Fgf13 splice isoforms could partially rescue deficits in interneuron action potential output and restore K+ channel current size.

Additionally, the discussion was nuanced, and demonstrated how the current findings resolved previous apparent contradictions in the field involving the function of FGF13.

These findings will have a significant impact on our understanding of how FGF13 causes seizures and death in DEEs, and the action of different FGF13 isoforms within different neuronal cell types, particularly GABAergic interneurons.

Comments on revisions:

I appreciate the author's responses to the previous round of reviews. All my comments have been addressed. Congratulations on an excellent body of work.

---

## [Author Response]

The following is the authors’ response to the original reviews.

**Public Reviews:**

**Reviewer #1 (Public Review):**
Summary:A subset of fibroblast growth factor (FGF) proteins (FGF11-FGF14; often referred to as fibroblast growth factor homologous factors because they are not thought to be secreted and do not seem to act as growth factors) have been implicated in modulating neuronal excitability, however, the exact mechanisms are unclear. In part, this is because it is unclear how different FGF isoforms alter ion channel activity in different neuronal populations. In this study, the authors explore the role of FGF 13 in epilepsy using a variety of FGF13 knock-out mouse models, including several targeted cell-type specific conditional knockout mouse lines. The study is intriguing as it indicates that FGF13 plays an especially important role in inhibitory neurons. Furthermore, although FGF13 has been studied as a regulator of neuronal voltage-gated sodium channels, the authors present data indicating that FGF13 knockout in inhibitory neurons induces seizures not by altering sodium current properties but by reducing voltage-gated potassium currents in inhibitory neurons. While intriguing, the data are incomplete in several aspects and thus the mechanisms by which various FGF13 variants induce Developmental and Epileptic Encephalopathies are not resolved by the data presented.Strengths:A major strength is the array of techniques used to assess the mice and the electrical activity of the neurons.The multiple mouse knock-out models utilized are a strength, clearly demonstrating that FGF13 expression in inhibitory neurons, and possibly specific sub-populations of inhibitory neurons, is critically important.The data on the increased sensitivity to febrile seizures in KO mice are very nice, provide clear evidence for regulation of excitability in inhibitory neurons by FGF13.The Gad2Fgf13-KO mice indicated that several Fgf13 splice variants may be expressed in inhibitory neurons and suggest that the Fgf13-VY splice variants may have previously unrecognized specific roles in regulating neuronal excitability.The data on males and females from the various KO mice lines indicates a clear gene dosage effect for this X-linked gene.The unbiased metabolomic analysis supports the assertion that Fgf13 expression in inhibitory neurons is important in regulating seizure susceptibility.Weaknesses:The knockout approach can be powerful but also has distinct limitations. Multiple missense mutations in FGF13-S have been identified. The knockout models employed here are not appropriate for understanding how these missense variants lead to altered neuronal excitability. While the data show that complete loss of Fgf13 from excitatory forebrain neurons is not sufficient to induce seizure susceptibility, it does not rule out that specific variants (e.g., R11C) might alter the excitability of forebrain neurons. The missense variants may alter excitatory and/or inhibitory neuron excitability in distinct ways from a full FGF13 knockout.

We agree with this overall interpretation of our data and have updated our language in the Discussion to make the distinction between mechanisms attributable to a knockout compared to a missense variant. We note, however, that the proposed mechanism by which missense variants (e.g., R11C) drive seizures is through loss of long-term inactivation in excitatory neurons and our excitatory knockout model shows loss of long-term inactivation in excitatory neurons. Thus, our knockout model demonstrates that the mechanism(s) by which the missense variants alter neuronal excitability in excitatory neurons must exclude long-term inactivation, thereby providing some clarity regarding the proposed mechanism for those missense variants.

The electrophysiological experiments are intriguing but not comprehensive enough to support all of the conclusions regarding how FGF13 modulates neuronal excitability.

We agree and have updated the language in our Discussion to clarify speculation from conclusions that are directly supported by data.

Another concern is the use of different ages of neurons for different experiments. For example, sodium currents in Figures 2 and 5 (and Supplemental Figures 2 and 7) are recorded from cultured neurons, which may have very different properties (including changes in sodium channel complexes) from neurons in vivo that drive the development of seizure activity.

We agree and acknowledge the important differences between neurons examined in culture and in vivo, yet the in vitro vs in vivo preparations were necessitated by the specific experiments. While these differences are important, previous gene profiling studies comparing primary hippocampal neurons with developing mouse hippocampus have found that although gene expression is accelerated in vitro, gene expression profiles in vitro and in vivo are similar (PMID: 11438693). Moreover, the relative immaturity of the cultured neurons is balanced at least in part because the in vivo experiments were performed on very young animals (~P12), which also have relatively immature neurons. Thus, we predict that sodium channel complexes studied in vitro are informative for the in vivo aspects of this investigation.

**Reviewer #2 (Public Review):**
Summary:The authors address three primary questions:(1) how FGF13 variants confer seizure susceptibility,(2) the specific cell types involved, and(3) the underlying mechanisms, particularly regarding Nav dysfunction.They use different Cre drivers to generate cell type-specific knockouts (KOs). First, using Nestin-Cre to create a whole-brain Fgf13 KO, they observed spontaneous seizures and premature death. While KO of Fgf13 in excitatory neurons does not lead to spontaneous seizures, KO in inhibitory neurons recapitulates the seizures and premature death observed in the Nestin-Cre KO. They further narrow down the critical cell type to MGE-derived interneurons (INs), demonstrating that MGE-neuron-specific KO partially reproduces the observed phenotypes. "All interneuron" KOs exhibit deficits in synaptic transmission and interneuron excitability, not seen in excitatory neuron-specific KOs. Finally, they rescue the defects in the interneuron-specific KO by expressing specific Fgf13 isoforms. This is an elegant and important study adding to our knowledge of mechanisms that contribute to seizures.Strengths• The study provides much-needed cell type-specific KO models.• The authors use appropriate Cre lines and characterize the phenotypes of the different KOs.• The metabolomic analysis complements the rest of the data effectively.• The study confirms and extends previous research using improved approaches (KO lines vs. in vitro KD or antibody infusion).• The methods and analyses are robust and well-executed.Weaknesses• One weakness lies in the use of the Nkx2.1 line (instead of Nkx2.1CreER) in the paper. As a result, some answers to key questions are incomplete. For instance, it remains unclear whether the observed effects are due to Chandelier cells or NGFCs, potentially both MGE and CGE derived, explaining why Nkx2.1 alone does not fully replicate the overall inhibitory KO. Using Nkx2.1CreER could have helped address the cell specificity. With the Nkx2.1 line used in the paper, the answer is partial.

We agree that while our data is consistent with the possibility of a role for Fgf13 in chandelier function, the current Cre driver does not provide sufficient direct evidence. We performed preliminary experiments (unpublished) using a Nkx2.1CreER driver, with late embryonic induction with a tamoxifen dosage validated for sparse labeling of chandelier cells (30846310). While we successfully replicated sparse labeling of neocortical chandelier cells (using a Cre-dependent Ai9 reporter), we were unable to determine if there was a significant loss of FGF13 as measured by immunohistochemistry since FGF13+ cells are only a small subset of the already sparse cells. Because multiple snRNA-seq studies identified Fgf13 as a marker for chandelier cells, we speculated—now more carefully circumspect—about the role of chandelier cells vs NGFCs.

• While the mechanism behind the reduced inhibitory drive in the IN-specific KO is suggested to be presynaptic, the chosen method does not allow them to exactly identify the mechanisms (spontaneous vs mEPSC/mIPSC), and whether it is a loss of inhibitory synapses (potentially axo-axonic) or release probability.

We agree that this is an important limitation of our work, and that we are unable to identify the exact mechanism behind the reduced inhibitory drive. We are continuing to explore this question in a follow-up study.

• Some supporting data (e.g. Supplemental Figure 7 and 8) appear to come from only one (or two) WT and one (or two) KO mice. Supplementary data, like main data, should come from at least three mice in total to be considered complete/solid (even if the statistical analysis is done with cells).

All panels in the manuscript, including supplementary data, except supplementary 7D and 8A, have N(mouse)≥3. Time limitations (graduating student) prevented us from obtaining a larger N. Because those supplementary data are not critical for supporting our conclusions, we removed them.

General AssessmentThe general conclusions of this paper are supported by data. As it is, the claim that "these results enhance our understanding of the molecular mechanisms that drive the pathogenesis of Fgf13-related seizures" is partially supported. A more cautious term may be more appropriate, as the study shows the mechanism is not Nav-mediated and suggests alternative mechanisms without unambiguously identifying them. The conclusion that the findings "expand our understanding of FGF13 functions in different neuron subsets" is supported, although somewhat overstated, as the work is not conclusive about the exact neuron subtypes. However, it does indeed show differential functions for specific neuronal classes, which is a significant result.Impact and UtilityThis paper is undoubtedly valuable. Understanding that excitatory neurons are not the primary contributors to the observed phenotypes is crucial. The finding that the effects are not MGE-unique is also important. This work provides a solid foundation for further research and will be a useful resource for future studies.
**Reviewer #3 (Public Review):**
Summary:The authors aimed to determine the mechanism by which seizures emerge in Developmental and Epileptic Encephalopathies caused by variants in the gene FGF13. Loss of FGF13 in excitatory neurons had no effect on seizure phenotype as compared to the loss of FGF13 in GABAergic interneurons, which in contrast caused a dramatic proseizure phenotype and early death in these animals. They were able to show that Fgf13 ablation and consequent loss of FGF13-S and FGF13-VY reduced overall inhibitory input from Fgf13-expressing interneurons onto hippocampal pyramidal neurons. This was shown to occur not via disruption to voltage-gated sodium channels but rather by reducing potassium currents and action potential repolarisation in these interneurons.Strengths:The authors employed multiple well-validated, novel mouse lines with FGF13 knocked out in specific cell types including all neurons, all excitatory cells, all GABAergic interneurons, or a subset of MGE-derived interneurons, including axo-axonic chandelier cells. The phenotypes of each of these four mouse lines were carefully characterised to reveal clear differences with the most fundamental being that Interneuron-targeted deletion of FGF13 led to perinatal mortality associated with extensive seizures and impaired the hippocampal inhibitory/excitatory balance while deletion of FGF13 in excitatory neurons caused no detectable seizures and no survival deficits.The authors made excellent use of western blotting and in situ hybridisation of the different FGF13 isoforms to determine which isoforms are expressed in which cell types, with FGF3-S predominantly in excitatory neurons and FGF13-VY and FGF13-V predominantly in GABAergic neurons.The authors performed a highly detailed electrophysiological analysis of excitatory neurons and GABAergic interneurons with FGF13 deficits using whole-cell patch clamp. This enabled them to show that FGF13 removal did not affect voltage-gated sodium channels in interneurons, but rather reduced the action of potassium channels, with the resultant effect of making it more likely that interneurons enter depolarisation block. These findings were strengthened by the demonstration that viral re-expression of different Fgf13 splice isoforms could partially rescue deficits in interneuron action potential output and restore K+ channel current size.Additionally, the discussion was nuanced and demonstrated how the current findings resolved previous apparent contradictions in the field involving the function of FGF13.These findings will have a significant impact on our understanding of how FGF13 causes seizures and death in DEEs, and the action of different FGF13 isoforms within different neuronal cell types, particularly GABAergic interneurons.
**Recommendations for the authors:**

**Reviewer #1 (Recommendations For The Authors):**
The limitations of the KO model should be fully discussed in the discussion. It should be clear that knocking out FGF13 does not provide insight into how missense mutations such as R11C may alter excitatory and/or inhibitory neuron excitability.

We agree with this overall interpretation of our data and have updated our language in the Discussion to make the distinction between mechanisms attributable to a knockout compared to a missense variant. We note, however, that the proposed mechanism by which missense variants (e.g., R11C) drive seizures is through loss of long-term inactivation in excitatory neurons and our excitatory knockout model shows loss of long-term inactivation in excitatory neurons. Thus, our knockout model demonstrates that the mechanism(s) by which the missense variants alter neuronal excitability in excitatory neurons must exclude long-term inactivation, thereby providing some clarity regarding the proposed mechanism for those missense variants.

It is important to know what sodium channel isoforms are expressed in the cultured neurons used in the experiments for Figures 2 and 5. Are Nav1.1, Nav1.2, Nav1.3, and Nav1.6 expressed at appropriate levels in the cultures?

We agree it is important to know that the sodium channel isoforms expressed in our hippocampal neurons are expressed at physiologically relevant levels, for further validation of our primary culture system. We have added RT-qPCR data from our hippocampal neuron cultures (Supplemental Figure 2B) showing the relative levels of SCN1A, SCN2A, SCN3A, and SCN8A, which are similar to the relative levels of voltage-gated sodium channel isoforms found in rodent and human forebrain in early development (Figure 1 in PMID: 35031483)*.*

The electrophysiological experiments are intriguing but limited. One, it would be helpful to report if there were any changes in resting membrane potential for the cells reported in Figure 5. It is also inappropriate to unequivocally state that "Nav currents were not significantly affected by Fgf13 knockout in Gad2Fghf13 KO neurons" as only a sampling of properties was investigated. Recovery from inactivation and persistent current amplitudes were not evaluated. Furthermore, while it looks like long-term inactivation is not altered, only one specific protocol was used and currents measured from cultured neurons may not be fully representative of neuronal properties in vivo.

We agree that we performed a selective analysis of Nav currents—selected because those are the major parameters that have been associated with FGF13 modulation. Because we did not observe significant differences in NaV currents, we therefore hypothesized that FGF13 affected other currents, as previously observed, and consequently assessed potassium currents, for which we did observe a difference. Further, we note that our sodium current and potassium current results are consistent with, and supportive of, our action potential data in which we find no deficit in AP initiation, but rather a deficit in AP repolarization. We revised the text to reflect the more limited analysis of Nav currents. Regarding long-term inactivation, we also agree that measurements in cultured neurons may not fully represent neuronal properties in vivo; however, we note that regulation of long-term inactivation by FGF13 has previously been assessed only in cultured cells (and not in neurons). Thus, our protocols were designed to query that modulation previously reported.

The first sentence of the results section is misleading: "To determine how FGF13 variants contribute to seizure disorders, we developed genetic mouse models that eliminate Fgf13 in specific neuronal cell types." The knockouts do not target specific splice isoforms and do not help determine how missense variants contribute to DEE. This should be modified to reflect better what is actually being tested.

We agree and have revised our text to state that our goal was to assess how FGF13 contributes to neuronal excitability and thereby accurately reflect the cell type-specific, but not isoform specific, targeting.

**Reviewer #2 (Recommendations For The Authors):**
• The sentence in the introduction stating "an unusual example of differential expression of an alternatively spliced neuronal gene in excitatory vs. inhibitor neurons" is factually incorrect, especially for transcripts regulating intrinsic properties like FGF13. Refer to PMID: 31451803 for more details and consider rephrasing this statement.

We updated our text to reflect the similarity of Fgf13’s cell type-specific alternative splicing to other genes known to control synaptic interactions and neuronal architecture and added the suggested reference.

• Consistency is needed in the manuscript regarding the term "BASEscope" or "basescope"; the correct version is "BaseScope."

We corrected the text accordingly.

• In the discussion, the term "reduced overall inhibitory drive" might be more appropriate than "input."

We updated the text accordingly.

• The authors should refer to the Fgf13 data in the database from Furlanis et al., which complements their findings: https://scheiffele-splice.scicore.unibas.ch/.

We agree and now incorporate this reference.

• The phrase "Fgf13 silencing in Nkx2.1 expressing neurons" should be clarified to include the use of CreER, which was crucial and effectively resulted in the labeling of a different subtype of interneurons, see PMID: 23180771.

We agree and have updated our text accordingly.

• Be more cautious when discussing the role of FGF13 in chandelier function; while it seems probable, the current Cre driver used provides no direct evidence.

We agree (as noted above) that while our data are consistent with the possibility of a role for Fgf13 in chandelier function, the current Cre driver used is insufficient to offer direct evidence and therefore updated our text in the discussion.

• The gene dosage effect is interesting, it would be interesting to explore it further in the future.

We agree. Because our data suggest that seizures result from loss of inhibitory neuron input, we hypothesize that the gene dosage effect derives from further loss of inhibitory neuron input and thus more hyperexcitability.

• Another critical aspect not addressed here and of interest for the future is the distinction between the role of FGF13 in interneuron development versus general maintenance. Using Nkx2.1CreER could have helped address both cell specificity and developmental roles.

We agree that there may be an interesting distinction between the role of Fgf13 in development versus general maintenance. We have piloted an Nkx2.1-CreER targeted deletion of Fgf13 from cortical interneurons but have been unsuccessful with significant deletion of Fgf13, likely because the Nkx2.1-CreER strategy targets only a sparse subset of interneurons and FGF13 is expressed in only a subset of total interneurons. Thus, use of the Nkxs.1-CreER strategy is challenging. We are looking for ways to optimize.

**Reviewer #3 (Recommendations For The Authors):**
This was a truly fabulous paper, with an exceptional quantity of beautiful data. I would like to congratulate the authors on their superb work.In the discussion, the authors correctly draw attention to the fact that the clear pro-seizure phenotype they see when FGF13 was knocked out more specifically in a subset of interneurons including chandelier cells, adds to our understanding of the role of FGF13 in chandelier cells. More than that though, given that FGF13 is reducing excitability in these cells AND this results in a strong pro-seizure phenotype, they may want to postulate that this lends further weight to the argument that chandeliers cells are likely powerful regulators of network excitability despite suggestions in the field that they could potentially have a proexcitatory function (see Szabadics et al. Science 2006).

We agree this is interesting and have elaborated on our discussion of chandelier cells to include this point while also addressing the important caveats noted by reviewer 2.

A minor point:On page 26 the sentence:"Here, we were able to assess FGF13-S and FGF13-VY, chosen because they are most abundantly expressed isoforms in the adult mouse brain, but the inability to rescue electrophysiological consequences completely with either isoform alone leaves open the possibility that other isoforms (e.g., FGF13-U, FGF13-V, and FGF13-VY) also make critical contributions." Should the last "FGF13-VY" be removed?

We thank the reviewer for noticing the error and have updated the text accordingly.